# A COMPRESSED SENSING VIEW OF UNSUPERVISED TEXT EMBEDDINGS, BAG-OF-$n$-GRAMS, AND LSTMS

**Sanjeev Arora, Mikhail Khodak, Nikunj Saunshi**
Princeton University
`{arora,mkhodak,nsaunshi}@cs.princeton.edu`

**Kiran Vodrahalli**
Columbia University
`kiran.vodrahalli@columbia.edu`

## ABSTRACT

Low-dimensional vector embeddings, computed using LSTMs or simpler techniques, are a popular approach for capturing the "meaning" of text and a form of unsupervised learning useful for downstream tasks. However, their power is not theoretically understood. The current paper derives formal understanding by looking at the subcase of linear embedding schemes. Using the theory of compressed sensing we show that representations combining the constituent word vectors are essentially information-preserving linear measurements of Bag-of-n-Grams (BonG) representations of text. This leads to a new theoretical result about LSTMs: low-dimensional embeddings derived from a low-memory LSTM are provably at least as powerful on classification tasks, up to small error, as a linear classifier over BonG vectors, a result that extensive empirical work has thus far been unable to show. Our experiments support these theoretical findings and establish strong, simple, and unsupervised baselines on standard benchmarks that in some cases are state of the art among word-level methods. We also show a surprising new property of embeddings such as GloVe and word2vec: they form a good sensing matrix for text that is more efficient than random matrices, the standard sparse recovery tool, which may explain why they lead to better representations in practice.

## 1 INTRODUCTION

Much attention has been paid to using LSTMs (Hochreiter & Schmidhuber, 1997) and similar models to compute text embeddings (Bengio et al., 2003; Collobert & Weston, 2008). Once trained, the LSTM can sweep once or twice through a given piece of text, process it using only limited memory, and output a vector with moderate dimensionality (a few hundred to a few thousand), which can be used to measure text similarity via cosine similarity or as a featurization for downstream tasks.

The powers and limitations of this method have not been formally established. For example, can such neural embeddings compete with and replace traditional linear classifiers trained on trivial Bag-of-$n$-Grams (BonG) representations? Tweaked versions of BonG classifiers are known to be a surprisingly powerful baseline (Wang & Manning, 2012) and have fast implementations (Joulin et al., 2017). They continue to give better performance on many downstream supervised tasks such as IMDB sentiment classification (Maas et al., 2011) than purely unsupervised LSTM representations (Kiros et al., 2015; Hill et al., 2016; Pagliardini et al., 2017). Even a very successful character-level (and thus computation-intensive, taking a month of training) approach does not reach BonG performance on datasets larger than IMDB (Radford et al., 2017). Meanwhile there is evidence suggesting that simpler *linear* schemes give compact representations that provide most of the benefits of word-level LSTM embeddings (Wieting et al., 2016; Arora et al., 2017). These linear schemes consist of simply adding up, with a few modifications, standard pretrained word embeddings such as GloVe or word2vec (Mikolov et al., 2013; Pennington et al., 2014).

The current paper ties these disparate threads together by giving an information-theoretic account of linear text embeddings. We describe linear schemes that preserve $n$-gram information as low-dimensional embeddings with *provable* guarantees for any text classification task. The previous linear schemes, which used unigram information, are subcases of our approach, but our best schemes can also capture $n$-gram information with low additional overhead. Furthermore, we show that the original unigram information can be (approximately) extracted from the low-dimensional embedding using sparse recovery/compressed sensing (Candès & Tao, 2005). Our approach also fits in the tradition of the older work on *distributed representations* of structured objects, especially the works of Plate (1995) and Kanerva (2009). The following are the main results achieved by this new world-view:

1. Using random vectors as word embeddings in our linear scheme (instead of pretrained vectors) already allows us to rigorously show that low-memory LSTMs are *provably* at least as good as every linear classifier operating on the full BonG vector. This is a novel theoretical result in deep learning, obtained relatively easily. By contrast, extensive empirical study of this issue has been inconclusive (apart from character-level models, and even then only on smaller datasets (Radford et al., 2017)). Note also that empirical work by its nature can only establish performance on some available datasets, not on *all* possible classification tasks. We prove this theorem in Section 4 by providing a nontrivial generalization of a result combining compressed sensing and learning (Calderbank et al., 2009). In fact, before our work we do not know of any provable quantification of the power of any text embedding.

2. We study theoretically and experimentally how our linear embedding scheme improves when it uses pretrained embeddings (GloVe etc.) instead of random vectors. Empirically we find that this improves the ability to preserve Bag-of-Words (BoW) information, which has the following restatement in the language of sparse recovery: word embeddings are better than random matrices for "sensing" BoW signals (see Section 5). We give some theoretical justification for this surprising finding using a new sparse recovery property characterizing when nonnegative signals can be reconstructed by $\ell_1$-minimization.

3. Section 6 provides empirical results supporting the above theoretical work, reporting accuracy of our linear schemes on multiple standard classification tasks. Our embeddings are consistently competitive with recent results and perform much better than all previous linear methods. Among unsupervised word-level representations they achieve state of the art performance on both the binary and fine-grained SST sentiment classification tasks (Socher et al., 2013). Since our document representations are fast, compositional, and simple to implement given standard word embeddings, they provide strong baselines for future work.

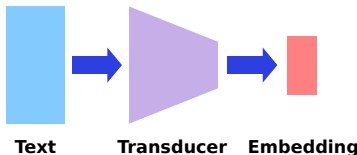

**Text**    **Transducer**  **Embedding**

## 2 RELATED WORK

Neural text embeddings are instances of *distributed representations*, long studied in connectionist approaches because they decay gracefully with noise and allow distributed processing. Hinton (1990) provided an early problem formulation, and Plate (1995) provided an elementary solution, the *holographic distributed representation*, which represents structured objects using circular vector convolution and has an easy and more compact implementation using the fast Fourier transform (FFT). Plate suggested applying such ideas to text, where "structure" can be quantified using parse trees and other graph structures. Our method is also closely related in form and composition to the sparse distributed memory system of Kanerva (2009). In the unigram case our embedding reduces to the familiar sum of word embeddings, which is known to be surprisingly powerful (Wieting et al., 2016), and with a few modifications even more so (Arora et al., 2017).

Representations of BonG vectors have been studied through the lens of compression by Paskov et al. (2013), who computed representations based on classical lossless compression algorithms using a linear program (LP). Their embeddings are still high-dimensional ($d > 100K$) and quite complicated to implement. In contrast, linear projection schemes are simpler, more compact, and can leverage readily available word embeddings. Pagliardini et al. (2017) also used a linear scheme, representing documents as an average of learned word and bigram embeddings. However, the motivation and benefits of encoding BonGs in low-dimensions are not made explicit. The novelty in the current paper is the connection to compressed sensing, which is concerned with recovering high-dimensional sparse signals $x \in \mathbb{R}^N$ from low-dimensional linear measurements $Ax$, specifically by studying conditions on matrix $A \in \mathbb{R}^{d \times N}$ when this is possible (see Appendix A for some background on compressed sensing and the previous work of Calderbank et al. (2009) that we build upon).

## 3 DOCUMENT EMBEDDINGS

In this section we define the two types of representations that our analysis will relate:

1. high-dimensional BonG vectors counting the occurrences of each $k$-gram for $k \leq n$
2. low-dimensional embeddings, from simple vector sums to novel $n$-gram-based embeddings

Although some of these representations have been previously studied and used, we define them so as to make clear their connection via compressed sensing, i.e. that representations of the second type are simply linear measurements of the first.

We now define some notation. Let $V$ be the number of words in the vocabulary and $V_n$ be the number of $n$-grams (independent of word order), so that $V = V_1$. Furthermore set $V_n^{\text{sum}} = \sum_{k \leq n} V_k$ and $V_n^{\text{max}} = \max_{k \leq n} V_k$. We will use words/$n$-grams and indices interchangeably, e.g. if $(a, b)$ is the $i$th of $V_2$ bigrams then the one-hot vector $e_{(a,b)}$ will be 1 at index $i$. Where necessary we will use $\{,\}$ to denote a multi-set and $(,)$ to denote a tuple. For any $m$ vectors $v_i \in \mathbb{R}^d$ for $i = 1, \ldots, m$ we define $[v_1, \ldots, v_m]$ to be their concatenation, which is thus an element of $\mathbb{R}^{md}$. Finally, for any subset $\mathcal{X} \subset \mathbb{R}^N$ we denote by $\Delta \mathcal{X}$ the set $\{x - x' : x, x' \in \mathcal{X}\}$.

### 3.1 THE BAG-OF-$n$-GRAMS VECTOR

Assigning to each word a unique index $i \in [V]$ we define the *Bag-of-Words* (BoW) representation $x^{\text{BoW}}$ of a document to be the $V$-dimensional vector whose $i$th entry is the number of times word $i$ occurs in the document. The $n$-gram extension of BoW is the *Bag-of-$n$-Grams* (BonG) representation, which counts the number of times any $k$-gram for $k \leq n$ appears in a document. Linear classification over such vectors has been found to be a strong baseline (Wang & Manning, 2012).

For ease of analysis we simplify the BonG approach by merging all $n$-grams in the vocabulary that contain the same words but in a different order. We call these features $n$-*cooccurrences* and find that the modification does not affect performance significantly (see Table 3 in Appendix F.1). Formally for a document $w_1, \ldots, w_T$ we define the *Bag-of-$n$-Cooccurrences* (BonC) vector as the concatenation

$$x^{\text{BonC}} = \left[ \sum_{t=1}^{T} e_{w_t}, \quad \cdots \quad , \quad \sum_{t=1}^{T-n+1} e_{\{w_t, \ldots, w_{t+n-1}\}} \right] \tag{1}$$

which is thus a $V_n^{\text{sum}}$-dimensional vector. Note that for unigrams this is equivalent to the BoW vector.

### 3.2 LOW-DIMENSIONAL $n$-GRAM EMBEDDINGS

Now suppose each word $w$ has a vector $v_w \in \mathbb{R}^d$ for some $d \ll V$. Then given a document $w_1, \ldots, w_T$ we define its *unigram embedding* as $z^{\text{u}} = \sum_{t=1}^{T} v_{w_t}$. While this is a simple and widely used featurization, we focus on the following straightforward relation with BoW: if $A \in \mathbb{R}^{d \times V}$ is a matrix whose columns are word vectors $v_w$ then $Ax^{\text{BoW}} = \sum_{t=1}^{T} Ae_{w_t} = \sum_{t=1}^{T} v_{w_t} = z^{\text{u}}$. Thus in terms of compressed sensing the unigram embedding of a document is a $d$-dimensional linear measurement of its Bag-of-Words vector.

We could extend this unigram embedding to $n$-grams by first defining a representation for each $n$-gram as the tensor product of the vectors of its constituent words. Thus for each bigram $b = (w_1, w_2)$ we would have $v_b = v_{w_1} v_{w_2}^T$ and more generally $v_g = \bigotimes_{t=1}^n v_{w_t}$ for each $n$-gram $g = (w_1, \ldots, w_n)$. The document embedding would then be the sum of the tensor representations of all $n$-grams.

The major drawback of this approach is of course the blowup in dimension, which in practice prevents its use beyond $n = 2$. To combat this a low-dimensional sketch or projection of the tensor product can be used, such as the circular convolution operator of Plate (1995). Since we are interested in representations that can also be constructed by an LSTM, we instead sketch this tensor product using the element-wise multiplication operation, which we find also usually works better than circular convolution in practice (see Table 4 in Appendix F.1). Thus for the $n$-cooccurrence $g = \{w_1, \ldots, w_n\}$, we define the *distributed cooccurrence* (DisC) embedding $\tilde{v}_g = d^{\frac{n-1}{2}} \bigodot_{t=1}^n v_{w_t}$. The coefficient is required when the vectors $v_w$ are random and unit norm to ensure that the product also has close to unit norm (see Lemma B.1). In addition to their convenient form, DisC embeddings have nice theoretical and practical properties: they preserve the original embedding dimension, they reduce to unigram (word) embeddings for $n = 1$, and under mild assumptions they satisfy useful compressed sensing properties with overwhelming probability (Lemma 4.1).

We then define the DisC document embedding to be the $nd$-dimensional weighted concatenation, over $k \leq n$, of the sum of the DisC vectors of all $k$-grams in a document:

$$z^{(n)} = \left[ C_1 \sum_{t=1}^T \tilde{v}_{w_t} , \quad \cdots \quad , \quad C_n \sum_{t=1}^{T-n+1} \tilde{v}_{\{w_t, \ldots, w_{t+n-1}\}} \right] \tag{2}$$

Here scaling factors $C_k$ are set so that all spans of $d$ coordinates have roughly equal norm (for random embeddings $C_k = 1$; for word embeddings $C_k = 1/k$ works well). Note that since $\tilde{v}_{w_t} = v_{w_t}$ we have $z^{(1)} = z^{\mathrm{u}}$ in the unigram case. Furthermore, as with unigram embeddings by comparing (1) and (2) one can easily construct a $\sum_{k=1}^n dn \times V_n^{\mathrm{sum}}$ matrix $A^{(n)}$ such that $z^{(n)} = A^{(n)} x^{\mathrm{BonC}}$.

## 3.3 LSTM REPRESENTATIONS

As discussed previously, LSTMs have become a common way to apply the expressive power of RNNs, with success on a variety of classification, representation, and sequence-to-sequence tasks. For document representation, starting with $h_0 = \mathbf{0}_m$ an $m$-*memory LSTM initialized with word vectors* $v_w \in \mathbb{R}^d$ takes in words $w_1, \ldots, w_T$ one-by-one and computes the document representation

$$h_t = f(\mathcal{T}_f(v_{w_t}, h_{t-1})) \circ h_{t-1} + i(\mathcal{T}_i(v_{w_t}, h_{t-1})) \circ g(\mathcal{T}_g(v_{w_t}, h_{t-1})) \tag{3}$$

where $h_t \in \mathbb{R}^m$ is the *hidden representation at time* $t$, the *forget gate* $f$, *input gate* $i$, and *input function* $g$ are a.e. differentiable nondecreasing elementwise "activation" functions $\mathbb{R}^m \mapsto \mathbb{R}^m$, and affine transformations $\mathcal{T}_*(x, y) = W_* x + U_* y + b_*$ have weight matrices $W_* \in \mathbb{R}^{m \times d}, U_* \in \mathbb{R}^{m \times m}$ and bias vectors $b_* \in \mathbb{R}^m$. The *LSTM representation* of a document is then the state at the last time step, i.e. $z^{\mathrm{LSTM}} = h_T$. Note that we will follow the convention of using *LSTM memory* to refer to the dimensionality of the hidden states. Since the LSTM is initialized with an embedding for each word it requires $\mathcal{O}(m^2 + md + Vd)$ computer memory, but the last term is just a lookup table so the vocabulary size does not factor into iteration or representation complexity.

From our description of LSTMs it is intuitive to see that one can initialize the gates and input functions so as to construct the DisC embeddings defined in the previous section. We state this formally and give the proof in the unigram case (the full proof appears in Appendix B.3):

**Proposition 3.1.** *Given word vectors $v_w \in \mathbb{R}^d$, one can initialize an $\mathcal{O}(nd)$-memory LSTM (3) that takes in words $w_1, \ldots, w_T$ (padded by an end-of-document token assigned vector $\mathbf{0}_d$) and constructs the DisC embedding (2) (up to zero padding), i.e. such that for all documents $z^{LSTM} = z^{(n)}$.*

*Proof (Unigram Case).* Set $f(x) = i(x) = g(x) = x$, $\mathcal{T}_f(v_{w_t}, h_{t-1}) = \mathcal{T}_i(v_{w_t}, h_{t-1}) = \mathbf{1}_d$, and $\mathcal{T}_g(v_{w_t}, h_{t-1}) = C_1 v_{w_t}$. Then $h_t = h_{t-1} + C_1 v_{w_t}$, so since $h_0 = \mathbf{0}_d$ we have the final LSTM representation $z^{\mathrm{LSTM}} = h_T = C_1 \sum_{t=1}^t v_{w_t} = z^{(1)}$. $\qquad \square$

By Proposition 3.1 we can construct a fixed LSTM that can compute compressed BonC representations on the fly and be further trained by stochastic gradient descent using the same memory.

## 4   LSTMs as Compressed Learners

Our main contribution is to provide the first rigorous analysis of the performance of the text embeddings that we are aware of, showing that the embeddings of Section 3.2 can provide performance on downstream classification tasks at least as well any linear classifier over BonCs. Before stating the theorem we make two mild simplifying assumptions on the BonC vectors:

1. The vectors are scaled by $\frac{1}{T\sqrt{n}}$, where $T$ is the maximum document length. This assumption is made without loss of generality.

2. No $n$-cooccurrence contains a word more than once. While this is (infrequently) violated in practice, the problem can be circumvented by merging words as a preprocessing step.

**Theorem 4.1.** *Let $S = \{(x_i, y_i)\}_{i=1}^m$ be drawn i.i.d. from a distribution $\mathcal{D}$ over BonC vectors of documents of length at most $T$ satisfying assumptions 1 and 2 above and let $w_0$ be the linear classifier minimizing the logistic loss $\ell_{\mathcal{D}}$. Then for dimension $d = \tilde{\Omega}\left(\frac{T}{\varepsilon^2}\log\frac{nV_n^{max}}{\gamma}\right)$ and appropriate choice of regularization coefficient one can initialize an $\mathcal{O}(nd)$-memory LSTM over i.i.d. word embeddings $v_w \sim \mathcal{U}^d\{\pm 1/\sqrt{d}\}$ such that w.p. $(1-\gamma)(1-2\delta)$ the classifier $\hat{w}$ minimizing the $\ell_2$-regularized logistic loss over its representations satisfies*

$$\ell_{\mathcal{D}}(\hat{w}) \leq \ell_{\mathcal{D}}(w_0) + \mathcal{O}\left(\|w_0\|_2\sqrt{\varepsilon + \frac{1}{m}\log\frac{1}{\delta}}\right) \tag{4}$$

The above theoretical bound shows that LSTMs match BonC performance as $\varepsilon \to 0$, which can be realized by increasing the embedding dimension $d$ (c.f. Figure 5).

### 4.1   Compressed Sensing and Learning

Compressed sensing is concerned with recovering a high-dimensional $k$-sparse signal $x \in \mathbb{R}^N$ from a few linear measurements; given a design matrix $A \in \mathbb{R}^{d\times N}$ this is formulated as

$$\text{minimize} \quad \|w\|_0 \quad \text{subject to} \quad Aw = z \tag{5}$$

where $z = Ax$ is the *measurement vector*. As $l_0$-minimization is NP-hard, research has focused on sufficient conditions for tractable recovery. One such condition is the *Restricted Isometry Property* (RIP), for which Candès & Tao (2005) proved that (5) can be solved by convex relaxation:

**Definition 4.1.** *$A \in \mathbb{R}^{d\times N}$ is $(\mathcal{X}, \varepsilon)$-RIP for some subset $\mathcal{X} \subset \mathbb{R}^N$ if $\forall\, x \in \mathcal{X}$*

$$(1-\varepsilon)\|x\|_2 \leq \|Ax\|_2 \leq (1+\varepsilon)\|x\|_2 \tag{6}$$

*We will abuse notation and say $(k, \varepsilon)$-RIP when $\mathcal{X}$ is the set of $k$-sparse vectors. This is the more common definition, but ours allows a more general Theorem 4.2 and a tighter bound in Theorem 4.1.*

Following these breakthroughs, Calderbank et al. (2009) studied whether it is possible to use the low-dimensional output of compressed sensing as a surrogate representation for classification. They proved a learning-theoretic bound on the loss of an SVM classifier in the compressed domain compared to the best classifier in the original domain. In this work we are interested in comparing the performance of LSTMs with BonC representations, so we need to generalize the Calderbank et al. (2009) result to handle Lipschitz losses and an arbitrary set $\mathcal{X} \subset \mathbb{R}^N$ of high-dimensional signals:

**Theorem 4.2.** *For any subset $\mathcal{X} \subset \mathbb{R}^N$ containing the origin let $A \in \mathbb{R}^{d\times N}$ be $(\Delta\mathcal{X}, \varepsilon)$-RIP and let $m$ samples $S = \{(x_i, y_i)\}_{i=1}^m \subset \mathcal{X} \times \{-1, 1\}$ be drawn i.i.d. from some distribution $\mathcal{D}$ over $\mathcal{X}$ with $\|x\|_2 \leq R$. If $\ell$ is a $\lambda$-Lipschitz convex loss function and $w_0 \in \mathbb{R}^N$ is its minimizer over $\mathcal{D}$ then w.p. $1 - 2\delta$ the linear classifier $\hat{w}_A \in \mathbb{R}^d$ minimizing the $\ell_2$-regularized empirical loss function $\ell_{S_A}(w) + \frac{1}{2C}\|w\|_2^2$ over the compressed sample $S_A = \{(Ax_i, y_i)\}_{i=1}^m \subset \mathbb{R}^d \times \{-1, 1\}$ satisfies*

$$\ell_{\mathcal{D}}(\hat{w}_A) \leq \ell_{\mathcal{D}}(w_0) + \mathcal{O}\left(\lambda R\|w_0\|_2\sqrt{\varepsilon + \frac{1}{m}\log\frac{1}{\delta}}\right) \tag{7}$$

*for appropriate choice of $C$. Recall that $\Delta\mathcal{X} = \{x - x' : x, x' \in \mathcal{X}\}$ for any $\mathcal{X} \subset \mathbb{R}^N$.*

While a detailed proof of this theorem is spelled out in Appendix C, the main idea is to compare the distributional loss incurred by a classifier $\hat{w}$ in the original space to the loss incurred by $A\hat{w}$ in the compressed space. We show that the minimizer of the regularized empirical loss in the original space ($\hat{w}$) is a bounded-coefficient linear combination of samples in $S$, so its loss depends only on inner products between points in $\mathcal{X}$. Thus using RIP and a generalization error result by Sridharan et al. (2008) we can bound the loss of $\hat{w}_A$, the regularized classifier in the compressed domain. Note that to get back from Theorem 4.2 the $\mathcal{O}(\sqrt{\varepsilon})$ bound for $k$-sparse inputs of Calderbank et al. (2009) we can set $\mathcal{X}$ to the be the set of $k$-sparse vectors and assume $A$ is $(2k, \varepsilon)$-RIP.

## 4.2 Proof of Main Result

To apply Theorem 4.2 we need the design matrix $A^{(n)}$ transforming BonCs into the DisC embeddings of Section 3.2 to satisfy the following RIP condition (Lemma 4.1), which we prove using a restricted isometry result for structured random sampling matrices in Appendix D:

**Lemma 4.1.** *Assume the setting of Theorem 4.1 and let $A^{(n)}$ be the $nd \times V_n^{sum}$ matrix relating DisC and BonC representations of any document by $z^{(n)} = A^{(n)} x^{BonC}$. If $d = \tilde{\Omega}\left(\frac{T}{\varepsilon^2} \log \frac{nV_n^{max}}{\gamma}\right)$ then $A^{(n)}$ is $\left(\Delta \mathcal{X}_T^{(n)}, \varepsilon\right)$-RIP w.p. $1 - \gamma$, where $\mathcal{X}_T^{(n)}$ is the set of BonCs of documents of length at most $T$.*

*Proof of Theorem 4.1.* Let $\hat{S} = \{(A^{(n)} x_i, y_i) : (x_i, y_i) \in S\}$, where $A^{(n)}$ is as in Lemma 4.1. Then by the same lemma $A^{(n)}$ is $\left(\Delta \mathcal{X}_T^{(n)}, \varepsilon\right)$-RIP w.p. $1 - \gamma$, where $\mathcal{X}_T^{(n)}$ is the set of BonC vectors of documents of length at most $T$. By BonC assumption (1) all BonCs lie within the unit ball, so we can apply Theorem 4.2 with $\ell$ the logistic loss, $\lambda = 1$, and $R = 1$ to get that a classifier $\hat{w}$ trained using $\ell_2$-regularized logistic loss over $\hat{S}$ will satisfy the required bound (4). Since by Proposition 3.1 one can initialize an $\mathcal{O}(nd)$-memory LSTM that takes in i.i.d. Rademacher word vectors $v_w \sim \mathcal{U}^d\{\pm 1/\sqrt{d}\}$ such that $z^{\text{LSTM}} = z^{(n)} = A^{(n)} x \ \forall \ x \in \mathcal{X}_T^{(n)}$, this completes the proof. $\qquad\square$

## 5 Sparse Recovery with Pretrained Embeddings

Theorem 4.1 is proved using random vectors as the word embeddings in the scheme of Section 3. However, in practice LSTMs are often initialized with standard word vectors such as GloVe. Such embeddings *cannot* satisfy traditional compressed sensing properties such as RIP or incoherence. This follows essentially from the definition: word embeddings seek to capture word similarity, so similar words (e.g. synonyms) have embeddings with high inner product, which violates both properties. Thus the efficacy of real-life LSTMs must have some other explanation. But in this section we present the surprising empirical finding that pretrained word embeddings are *more efficient* than random vectors at encoding and recovering BoW information via compressed sensing. We further sketch a potential explanation for this result, though a rigorous explanation is left for subsequent work.

### 5.1 Pretrained Embeddings Preserve Sparse Information

In recent years word embeddings have been discovered to have many remarkable properties, most famously the ability to solve analogies (Mikolov et al., 2013). Our connection to compressed sensing indicates that they should have another: preservation of sparse signals as low-dimensional linear measurements. To examine this we subsample documents from the SST (Socher et al., 2013) and IMDB (Maas et al., 2011) classification datasets, embed them as $d$-dimensional unigram embeddings $z = Ax$ for $d = 50, 100, 200, \ldots, 1600$ (where $A \in \mathbb{R}^{d \times V}$ is the matrix of word embeddings and $x$ is a document's BoW vector), solve the following LP, known as *Basis Pursuit* (BP), which is the standard $\ell_1$-minimization problem for sparse recovery in the noiseless case (see Appendix A):

$$\text{minimize} \quad \|w\|_1 \quad \text{subject to} \quad Aw = z \tag{8}$$

Success is measured as the $F_1$ score of retrieved words. We use Squared Norm (SN) vectors (Arora et al., 2016) trained on a corpus of Amazon reviews (McAuley et al., 2015) and normalized i.i.d. Rademacher vectors as a baseline. SN is used due to similarity to GloVe and its formulation via an easy-to-analyze generative model that may provide a framework to understand the results (see Appendix F.2), while the Amazon corpus is used for its semantic closeness to the sentiment datasets.

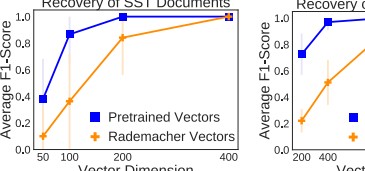 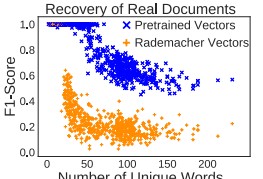 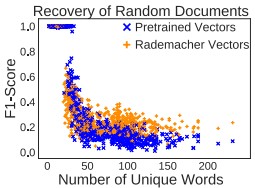

Figure 1: Average $F_1$-score of 200 recovered BoW vectors from SST (left) and IMDB (right) compared to dimension. Pretrained word embeddings (SN trained on Amazon reviews) need half the dimensionality of normalized Rademacher vectors to achieve near-perfect recovery. Note that IMDB documents are on average more than ten times longer than SST documents.

Figure 2: $F_1$-score of 1000 recovered BoWs compared to number of unique words. Real documents (left) are drawn from the SST and IMDB corpora; random signals (right) are created by picking words at random. For $d = 200$, pretrained embeddings are better than Rademacher vectors as sensing vectors for natural language BoW but are worse for random sparse signals.

Figures 1 and 2 show that pretrained embeddings require a lower dimension $d$ than random vectors to recover natural language BoW. This is surprising as the training objective goes against standard conditions such as approximate isometry and incoherence; indeed as shown in Figure 2 recovery is poor for randomly generated word collections. The latter outcome indicates that the fact that a document is a set of mutually meaningful words is important for sparse recovery using embeddings trained on co-occurrences. We achieve similar results with other objectives (e.g. GloVe/word2vec) and other corpora (see Appendix F.1), although there is some sensitivity to the sparse recovery method, as other $\ell_1$-minimization methods work well but greedy methods, such as Orthogonal Matching Pursuit (OMP), work poorly, likely due to their dependence on incoherence (Tropp, 2004).

For the $n$-gram case (i.e. BonC recovery for $n > 1$), although we know by Lemma 4.1 that DisC embeddings composed from random vectors satisfy RIP, for pretrained vectors it is unclear how to reason about suitable $n$-gram embeddings without a rigorous understanding of the unigram case, and experiments do not show the same recovery benefits. One could perhaps do well by training on cooccurrences of word tuples, but such embeddings could not be used by a word-level LSTM.

## 5.2 Understanding Sparse Recovery of Natural Language Documents

As shown in Figure 2, the success of pretrained embeddings for linear sensing is a local phenomenon; recovery is only efficient for naturally occurring collections of words. However, applying statistical RIP/incoherence ideas (Barg et al., 2015) to explain this is ruled out since they require collections to be incoherent with high probability, whereas word embeddings are trained to give high inner product to words appearing together. Thus an explanation must come from some other, weaker condition. The usual necessary and sufficient requirement for recovering all signals with support $S \subset [N]$ is the *local nullspace property* (NSP), which stipulates that vectors in the kernel of $A$ not have too much mass on $S$ (see Definition A.2). While NSP and related properties such as *restricted eigenvalue* (see Definition A.3) are hard to check, we can impose some additional structure to formulate an intuitive, verifiable perfect recovery condition for our setting. Specifically, since our signals (BoW vectors) are nonnegative, we can improve upon solving BP (8) by instead solving *nonnegative BP* (BP+):

$$\text{minimize} \quad \|w\|_1 \quad \text{subject to} \quad Aw = z, \ w \geq \mathbf{0}_d \tag{9}$$

The following geometric result then characterizes when solutions of BP+ recover the correct signal:

**Theorem 5.1** (Donoho & Tanner, 2005)**.** *Consider a matrix $A \in \mathbb{R}^{d \times N}$ and an index subset $S \subset [N]$ of size $k$. Then any nonnegative vector $x \in \mathbb{R}_+^N$ with support $\operatorname{supp}(x) = S$ is recovered from $Ax$ by BP+ iff the set $A_S$ of columns of $A$ indexed by $S$ comprise the vertices of a $k$-dimensional face of the convex hull $\operatorname{conv}(A)$ of the columns of $A$ together with the origin.*

This theorem equates perfect recovery of a BoW vector via BP+ with the vectors of its words being the vertices of some face of the polytope $\operatorname{conv}(A)$. The property holds for incoherent columns since the vectors are far enough that no one vector is inside the simplex formed by any $k$ others. On the other hand, pretrained embeddings satisfy it by having commonly co-occurring words close together and other words far away, making it easier to form a face from columns indexed by the support of a BoW. We formalize this intuition as the *Supporting Hyperplane Property* (SHP):

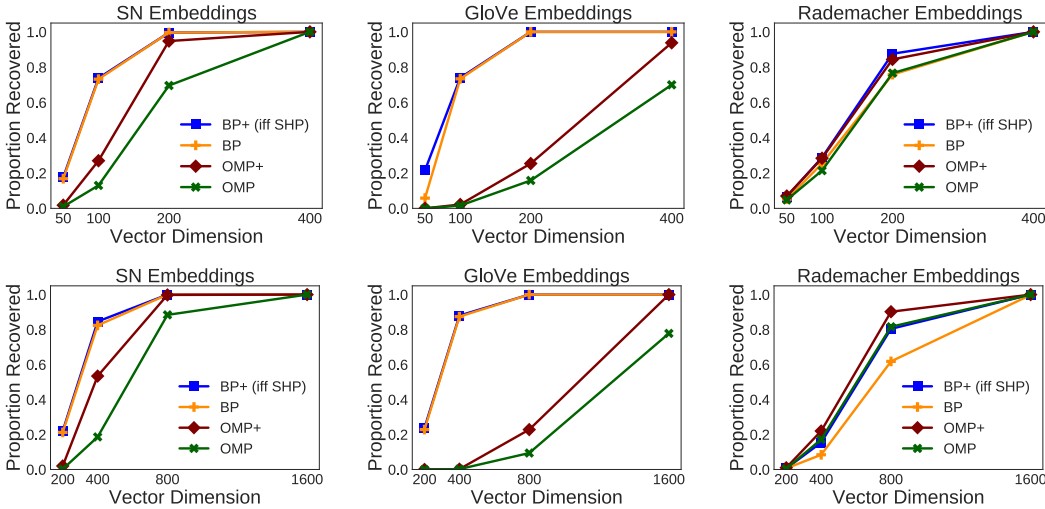

Figure 3: Proportion of 500 randomly sampled documents from SST (top) and IMDB (bottom) that are perfectly recovered from linear measurements.

**Definition 5.1.** *A matrix $A \in \mathbb{R}^{d \times N}$ satisfies S-SHP for subset $S \subset [N]$ if its columns are in general position and there is a hyperplane containing the set $A_S$ of columns of $A$ indexed by $S$ such that the set of all other columns of $A$ together with the origin are on one side of the hyperplane.*

SHP is a very weak property implied by NSP (Corollary E.1). However, it can be checked by using convex optimization to see if the hyperplane exists (Appendix E.2). Furthermore, we show (full proof in Appendix E.1) that this hyperplane is the *supporting hyperplane* of the face of $\mathrm{conv}(A)$ with vertices $A_S$, from which it follows by Theorem 5.1 that SHP *characterizes* recovery using BP+:

**Corollary 5.1.** *BP+ recovers any $x \in \mathbb{R}_+^N$ with $\mathrm{supp}(x) = S$ from $Ax$ iff $A$ satisfies $S$-SHP.*

*Proof Sketch.* By Theorem 5.1 it suffices to show equivalence of $S$-SHP with the column set $A_S$ comprising the vertices of a $k$-dimensional face of $\mathrm{conv}(A)$. A *face $F$* of polytope $P$ is defined as its intersection with some hyperplane such that all points in $P \backslash F$ lie on one side of the hyperplane.
( $\implies$ ) Let $F$ be the face of $\mathrm{conv}(A)$ formed by the columns $A_S$. Then there must be a supporting hyperplane $H$ containing $F$. Since the columns of $A$ are in general position, all columns $A_{\overline{S}} = A \backslash A_S$ lie in $\mathrm{conv}(A) \backslash F$ and hence must all be on one side of $H$, so $H$ is the desired hyperplane.
( $\impliedby$ ) Let $H$ be the hyperplane supporting $A_S$, with all other columns on one side of $H$. By convexity, $H$ contains the simplex $F$ of $A_S$. Any point in $\mathrm{conv}(A) \backslash F$ can be written as a convex combination of points in $F$ and columns $A_{\overline{S}}$, with a positive coefficient on at least one of the columns, and so must lie on the same side of $H$ as $A_{\overline{S}}$. Thus $A_S$ comprises the vertices of a face $F$. $\qquad\square$

Thus perfect recovery of a BoW via BP+ is equivalent to the existence of a hyperplane separating embeddings of words in the document from those of the rest of the vocabulary. Intuitively, words in the same document are trained to have similar embeddings and so will be easier to separate out, providing some justification for why pretrained vectors are better for sensing. We verify that SHP is indeed more likely to be satisfied by such designs in Figure 3, which also serves as an empirical check of Corollary 5.1 since SHP satisfaction implies BP recovery as the latter can do no better than BP+. We further compare to recovery using OMP/OMP+ (the latter removes negative values and recomputes the set of atoms at each iteration); interestingly, while OMP+ recovers the correct signal from SN almost as often as BP/BP+, it performs quite poorly for GloVe, indicating that these embeddings may have quite different sensing properties despite similar training objectives.

As similarity properties that may explain these results also relate to downstream task performance, we conjecture a relationship between embeddings, recovery, and classification that may be understood under a generative model (see Appendix F.2). However, the Section 4 bounds depend on RIP, not recovery, so these experiments by themselves do not apply. They do show that the compressed sensing framework remains relevant even in the case of non-random, pretrained word embeddings.

| Representation | $n$ | $d^*$ | MR | CR | SUBJ | MPQA | TREC | SST ($\pm 1$) | SST | IMDB |
|---|---|---|---|---|---|---|---|---|---|---|
| BonC (1) | 1 | $V_1$ | 77.1 | 77.0 | 91.0 | 85.1 | 86.8 | 80.7 | 36.8 | 88.3 |
| | 2 | $V_2^{\text{sum}}$ | 77.8 | 78.1 | 91.8 | 85.8 | 90.0 | 80.9 | 39.0 | **90.0** |
| | 3 | $V_3^{\text{sum}}$ | 77.8 | 78.3 | 91.4 | 85.6 | 89.8 | 80.1 | 42.3 | 89.8 |
| DisC (2) | 1 | 1600 | 79.6 | 81.0 | 92.4 | 87.8 | 85.2 | 84.6 | 45.7 | 89.2 |
| | 2 | 3200 | **80.1** | 81.5 | 92.6 | 87.9 | 89.6 | **85.5** | **46.4** | 89.4 |
| | 3 | 4800 | 80.0 | 81.3 | 92.6 | 87.9 | 90.0 | **85.2** | **46.7** | 89.6 |
| SIF[1] | 1 | 1600 | 79.6 | 81.1 | 92.5 | 87.7 | 85.6 | 84.4 | 45.8 | 89.2 |
| Sent2Vec[2] | 1 | 700 | 76.2 | 78.7 | 91.2 | 87.2 | 85.8 | 80.2 | 31.0 | 85.5 |
| Sent2Vec[2] | 2 | 700 | 76.3 | 79.1 | 91.1 | 86.6 | 84.2 | 80.0 | 30.7 | 85.3 |
| CFL[3] | 5 | 100K+ | | | | | | | | **90.4** |
| Paragraph Vec.[4] | | | 74.8 | 78.1 | 90.5 | 74.2 | **91.8** | | | |
| skip-thoughts[4] | | 4800 | **80.3** | **83.8** | **94.2** | **88.9** | **93.0** | 85.1 | 45.8 | |
| SDAE[5] | | 2400 | 74.6 | 78.0 | 90.8 | 86.9 | 78.4 | | | |
| CNN-LSTM[6] | | 4800 | 77.8 | **82.0** | **93.6** | **89.4** | 92.6 | | | |
| byte mLSTM[7] | | 4096 | *86.8* | *90.6* | *94.7* | *88.8* | 90.4 | *91.7* | *54.6* | *92.2* |

[*] Vocabulary sizes (i.e. BonC dimensions) vary by task; usually 10K-100K.
[1] Arora et al. (2017) Reported performance of best hyperparameter using Amazon GloVe embeddings.
[2,4,7] Pagliardini et al. (2017); Kiros et al. (2015); Radford et al. (2017) Evaluated latest pretrained models. Note that the available skip-thoughts implementation fails on the IMDB and MRPC tasks
[3,5,6] Paskov et al. (2013); Hill et al. (2016); Gan et al. (2017) From publication (+emb version of last two).

Table 1: Evaluation of DisC and recent unsupervised word-level approaches on standard classification tasks, with the character LSTM of Radford et al. (2017) shown for comparison. The top three results for each dataset are **bolded**, the best is *italicized*, and the best word-level performance is underlined.

## 6 EMPIRICAL FINDINGS

Our theoretical results show that simple tensor product sketch-based $n$-gram embeddings can approach BonG performance and be computed by a low-memory LSTM. In this section we compare these text representations and others on several standard tasks, verifying that DisC performance approaches that of BonCs as dimensionality increases and establishing several baselines for text classification. Code to reproduce results is provided at `https://github.com/NLPrinceton/text_embedding`.

**Tasks:** We test classification on MR movie reviews (Pang & Lee, 2005), CR customer reviews (Hu & Liu, 2004), SUBJ subjectivity dataset (Pang & Lee, 2004), MPQA opinion polarity subtask (Wiebe et al., 2005), TREC question classification (Li & Roth, 2002), SST sentiment classification (binary and fine-grained) (Socher et al., 2013), and IMDB movie reviews (Maas et al., 2011). The first four are evaluated using 10-fold cross-validation, while the others have train-test splits. In all cases we use logistic regression with $\ell_2$-regularization determined by cross-validation. We further test DisC on the SICK relatedness and entailment tasks (Marelli et al., 2014) and the MRPC paraphrase detection task (Dolan & Brockett, 2005). The inputs here are sentences pairs $(a, b)$ and the standard featurization for document embeddings $x_a$ and $x_b$ of $a$ and $b$ is $[|x_a - x_b|, x_a \odot x_b]$ (Tai et al., 2015). We use logistic regression for SICK entailment and MRPC and use ridge regression to predict similarity scores for SICK relatedness, with $\ell_2$-regularization determined by cross-validation. Since BonGs are not used for pairwise tasks our theory says nothing about performance here; we include these evaluations to show that our representations are also useful for other tasks.

**Embeddings:** In the main evaluation (Table 1) we use normalized 1600-dimensional GloVe embeddings (Pennington et al., 2014) trained on the Amazon Product Corpus (McAuley et al., 2015), which are released at `http://nlp.cs.princeton.edu/DisC`. We also compare the SN vectors of Section 5 trained on the same corpus with random vectors when varying the dimension (Figure 5).

| Rep. | $n$ | SICK-R $(r/\rho)$ | SICK-E | MRPC (Acc./$F_1$) |
|---|---|---|---|---|
| | 1 | 73.6 / 71.0 | 81.9 | 73.1 / **81.8** |
| DisC (2) | 2 | 75.6 / 72.1 | **83.2** | 70.8 / 79.0 |
| | 3 | **76.2** / **72.2** | **82.5** | 73.1 / 81.6 |
| SIF | 1 | 73.7 / 68.5 | 82.4 | 73.5 / **82.1** |
| Sent2Vec | 1 | 69.3 / 64.6 | 78.6 | 71.3 / 80.7 |
| Sent2Vec | 2 | 70.1 / 65.3 | 78.7 | 70.0 / 79.3 |
| skip-thoughts | | **82.4** / **76.0** | **83.2** | |
| SDAE | | | | 73.7 / 80.7 |
| CNN-LSTM | | | | **76.4** / **83.8** |
| byte mLSTM | | **78.5** / **72.1** | 80.0 | **73.8** / 81.4 |

Model information is the same as that in Table 1.

Table 2: Performance of DisC and other recent approaches on pairwise similarity and classification tasks. The top three results for each task are **bolded** and the best is underlined.

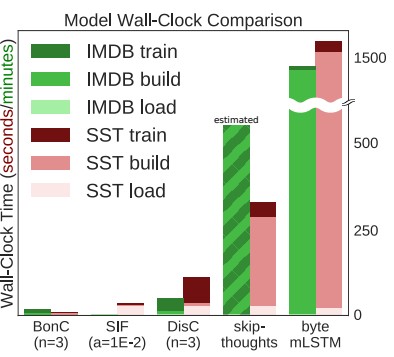

Figure 4: Time needed to initialize model, construct document representations, and train a linear classifier on a 16-core compute node.

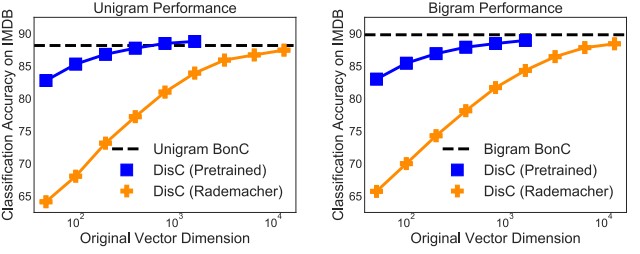

Figure 5: IMDB performance of unigram (left) and bigram (right) DisC embeddings compared to the original dimension.

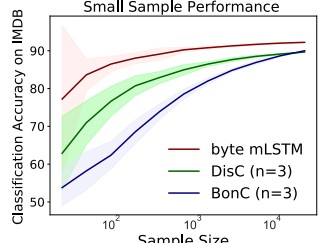

Figure 6: IMDB performance compared to training sample size.

**Results:** We find that DisC representation performs consistently well relative to recent unsupervised methods; among word-level approaches it is the top performer on the SST tasks and competes on many others with skip-thoughts and CNN-LSTM, both concatenations of two LSTM representations. While success may be explained by training on a large and in-domain corpus, being able to use so much text without extravagant computing resources is one of the advantages of a simple approach. Overall our method is useful as a strong baseline, often beating BonCs and many more complicated approaches while taking much less time to represent and train on documents than neural representations (Figure 4).

Finally, we analyze empirically how well our model approximates BonC performance. As predicted by Theorem 4.1, the performance of random embeddings on IMDB approaches that of BonC as dimension increases and the isometry distortion $\varepsilon$ decreases (Figure 5). Using pretrained (SN) vectors, DisC embeddings approach BonC performance much earlier, surpassing it in the unigram case.

# 7 CONCLUSION

In this paper we explored the connection between compressed sensing, learning, and natural language representation. We first related LSTM and BonG methods via word embeddings, coming up with simple new document embeddings based on tensor product sketches. Then we studied their classification performance, proving a generalization of the compressed learning result of Calderbank et al. (2009) to convex Lipschitz losses and a bound on the loss of a low-dimensional LSTM classifier in terms of its (modified) BonG counterpart, an issue which neither experiments nor theory have been able to resolve. Finally, we showed how pretrained embeddings fit into this sparse recovery framework, demonstrating and explaining their ability to efficiently preserve natural language information.

ACKNOWLEDGMENTS

We thank Rong Ge, Holden Lee, and Divyarthi Mohan for helpful discussions at various stages of this effort. The work in this paper was in part supported by NSF grants CCF-1302518 and CCF-1527371, Simons Investigator Award, Simons Collaboration Grant, and ONR-N00014-16-1-2329

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

# A   COMPRESSED SENSING BACKGROUND

The field of compressed sensing is concerned with recovering a high-dimensional $k$-sparse signal $x \in \mathbb{R}^N$ from few linear measurements. In the noiseless case this is formulated as

$$\text{minimize} \quad \|w\|_0 \quad \text{subject to} \quad Aw = z \tag{10}$$

where $A \in \mathbb{R}^{d \times N}$ is the *design matrix* and $z = Ax$ is the *measurement vector*. Since $\ell_0$-minimization is NP-hard, a foundational approach is to use its convex surrogate, the $\ell_1$-norm, and characterize when the solution to (10) is equivalent to that of the following LP, known as *basis pursuit* (BP):

$$\text{minimize} \quad \|w\|_1 \quad \text{subject to} \quad Aw = z \tag{11}$$

Related approaches such as *Basis Pursuit Denoising* (LASSO) and the *Dantzig Selector* generalize BP to handle signal or measurement noise (Foucart & Rauhut, 2013); however, the word embeddings case is noiseless so these methods reduce to BP. Note that throughout Section 5 and the Appendix we say that an $\ell_1$-minimization method *recovers $x$ from $Ax$* if its optimal solution is unique and equivalent to the optimal solution of (10).

An alternative way to approximately solve (10) is to use a greedy algorithm such as *matching pursuit* (MP) or *orthogonal matching pursuit* (OMP), which pick basis vectors one at a time by multiplying the measurement vector by $A^T$ and choosing the column with the largest inner product (Tropp, 2004).

## A.1   SOME COMMON SPARSE RECOVERY CONDITIONS

One condition through which recovery can be guaranteed is the *Restricted Isometry Property* (RIP):

**Definition A.1.** $A \in \mathbb{R}^{d \times N}$ *is* $(k, \varepsilon)$*-RIP if for all $k$-sparse* $x \in \mathbb{R}^N$

$$(1 - \varepsilon)\|x\|_2 \leq \|Ax\|_2 \leq (1 + \varepsilon)\|x\|_2$$

A line of work started by Candès & Tao (2005) used the RIP property to characterize matrices $A$ such that (10) and (11) have the same minimizer for any $k$-sparse signal $x$; this occurs with overwhelming probability when $d = \Omega\left(k \log \frac{N}{k}\right)$ and $\sqrt{d}A_{ij} \sim \mathcal{N}(0, 1) \,\forall\, i, j$ or $\sqrt{d}A_{ij} \sim \mathcal{U}\{-1, 1\} \,\forall\, i, j$.

Since the ability to recover a signal $x$ from a representation $Ax$ implies information preservation, a natural next step is to consider learning after compression. Calderbank et al. (2009) show that for $m$ i.i.d. $k$-sparse samples $\{(x_i, y_i)\}_{i=1}^m$ and a $(2k, \varepsilon)$-RIP matrix $A$, the hinge loss of a classifier trained on $\{(Ax_i, y_i)\}_{i=1}^m$ is bounded by that of the best linear classifier over the original samples. Theorem 4.2 provides a generalization of this result to any convex Lipschitz loss function.

RIP is a strong requirement, both because it is not necessary for perfect, stable recovery of $k$-sparse vectors using $\tilde{\mathcal{O}}(k)$ measurements and because in certain settings we are interested in using the above ideas to recover specific signals — those statistically likely to occur—rather than all $k$-sparse signals. The usual necessary and sufficient condition to recover any vector $x \in \mathbb{R}^N$ with index support set $S \subset [N]$ is the *local nullspace property* (NSP), which is implied by RIP:

**Definition A.2** (Foucart & Rauhut, 2013). *A matrix $A \in \mathbb{R}^{d \times N}$ satisfies NSP for a set $S \subset [N]$ if* $\|w_S\|_1 < \|w_{\overline{S}}\|_1$ *for all nonzero* $w \in \ker(A) = \{v : Av = \mathbf{0}_d\}$.

**Theorem A.1** (Foucart & Rauhut, 2013). *BP (11) recovers any $x \in \mathbb{R}_+^N$ with $\text{supp}(x) = S$ from $Ax$ iff $A$ satisfies NSP for $S$.*

A related condition that implies NSP is the *local restricted eigenvalue property* (REP):

**Definition A.3** (Raskutti et al., 2010). *A matrix $A \in \mathbb{R}^{d \times N}$ satisfies $\gamma$-REP for a set $S \subset [N]$ if $\|Aw\|_2 \geq \gamma \sqrt{d} \|w\|_2$ whenever $\|w_{\overline{S}}\|_1 \leq \|w_S\|_1$.*

Lastly, a simple condition that can sometimes provide recovery guarantees is *mutual incoherence*:

**Definition A.4.** *$A \in \mathbb{R}^{d \times N}$ is $\mu$-incoherent if $\max_{a,a'} |a^T a'| \leq \mu$, where the maximum is taken over any two distinct columns $a, a'$ of $A$.*

While incoherence is easy to verify (unlike the previous recovery properties), word embeddings tend to have high coherence due to the training objective pushing together vectors of co-occurring words.

## A.2 Conditions for Recovering Nonnegative Signals

Apart from incoherence, the properties above are hard show empirically. However, we are compressing BoW vectors, so our signals are nonnegative and we can impose an additional constraint on (11):

$$\text{minimize} \quad \|w\|_1 \quad \text{subject to} \quad Aw = z, \ w \geq \mathbf{0}_d \tag{12}$$

The following geometric result provides guarantees for this *nonnegative basis pursuit* (BP+) problem:

**Theorem A.2** (Donoho & Tanner, 2005). *Consider a matrix $A \in \mathbb{R}^{d \times N}$ and an index subset $S \subset [N]$ of size $k$. Then any nonnegative vector $x \in \mathbb{R}_+^N$ with support $\text{supp}(x) = S$ is recovered from $Ax$ by BP+ (12) iff the columns of $A$ indexed by $S$ comprise the vertices of a $k$-dimensional face of the convex hull $\text{conv}(A)$ of the columns of $A$ together with the origin.*

The polytope condition is equivalent to *nonnegative NSP* (NSP+), a weaker form of NSP:

**Definition A.5** (Foucart & Koslicki, 2014). *A matrix $A \in \mathbb{R}^{d \times N}$ satisfies NSP+ for a set $S \subset [N]$ if*

$$w_{\overline{S}} \geq \mathbf{0}_N \implies \sum_{i=1}^{N} w_i > 0 \text{ for all nonzero } w \in \ker(A).$$

**Lemma A.1.** *If $A \in \mathbb{R}^{d \times N}$ satisfies NSP for some $S \subset [N]$ then it also satisfies NSP+ for $S$.*

*Proof (Adapted from Foucart & Koslicki (2014)).* Since A satisfies NSP, we have $\|w_S\|_1 < \|w_{\overline{S}}\|_1$. Then for a nonzero $w \in \ker(A)$ such that $w_{\overline{S}} \geq \mathbf{0}$ we will have

$$\sum_{i=1}^{N} w_i = \sum_{i \in S} w_i + \sum_{j \in \overline{S}} w_j \geq -\sum_{i \in S} |w_i| + \sum_{j \in \overline{S}} |w_j| = -\|w_S\|_1 + \|w_{\overline{S}}\|_1 > 0$$

$\square$

**Lemma A.2.** *BP+ recovers any $x \in \mathbb{R}_+^N$ with $\text{supp}(x) = S$ from $Ax$ iff $A$ satisfies NSP+ for $S$.*

*Proof.* ( $\implies$ ): For any nonzero $w \in \ker(A)$ such that $w_{\overline{S}} \geq \mathbf{0}$, $\exists \lambda > 0$ such that $x + \lambda w \geq \mathbf{0}_N$ and $A(x + \lambda w) = Ax$. Since BP+ uniquely recovers $x$, we have $\|x + \lambda w\|_1 > \|x\|_1$, so NSP+ follows from the following inequality and the fact that $\lambda$ is positive:

$$0 < \|x + \lambda w\|_1 - \|x\|_1 = \sum_{i=1}^{N} (x_i + \lambda w_i) - \sum_{i=1}^{N} x_i = \lambda \sum_{i=1}^{N} w_i \implies \sum_{i=1}^{N} w_i > 0$$

( $\impliedby$ ): For any $x' \geq \mathbf{0}$ such that $Ax' = Ax$ we have that $w = x' - x \in \ker(A)$ and $w_{\overline{S}} = x'_{\overline{S}} \geq \mathbf{0}$ since the support of $x$ is $S$. Thus by NSP+ we have that $\sum_{i=1}^{N} w_i > 0$, which yields

$$\|x'\|_1 - \|x\|_1 = \sum_{i=1}^{N} x'_i - \sum_{i=1}^{N} x_i = \sum_{i=1}^{N} w_i > 0$$

Thus BP+ will recover $x$ uniquely. $\square$

Lemma A.2 shows that NSP+ is equivalent to the polytope condition in Theorem A.2, as they are both necessary and sufficient conditions for BP+ recovery.

| Model | Representation | MR | CR | SUBJ | MPQA | TREC | SST ($\pm 1$) | SST | IMDB |
|-------|----------------|------|------|------|------|------|------|------|------|
| Bigram | BonG | **78.3** | 78.2 | 91.9 | 85.7 | 90.0 | 80.8 | 39.1 | 90.0 |
|        | BonC | **77.8** | 78.1 | 91.8 | 85.8 | 90.0 | 80.9 | 39.0 | 90.0 |
| Trigram | BonG | 77.9 | 78.3 | 91.5 | 85.6 | 89.2 | 80.2 | **39.1** | 90.2 |
|         | BonC | 77.8 | 78.3 | 91.4 | 85.6 | 89.8 | 80.1 | **42.3** | 89.8 |

Table 3: The performance of an $l_2$-regularized logit classifier over Bag-of-$n$-Grams (BonG) vectors is generally similar to that of Bag-of-$n$-Cooccurrences (BonC) vectors for $n = 2, 3$ (largest differences **bolded**). Evaluation settings are the same as in Section 6. Note that for unigrams the two representations are equivalent.

| Model | Representation | MR | CR | SUBJ | MPQA | TREC | SST ($\pm 1$) | SST | IMDB |
|-------|----------------|------|------|------|------|------|------|------|------|
| Bigram | DisC (2) | **80.1** | 81.5 | 92.6 | 87.9 | 89.6 | **85.5** | 46.4 | 89.4 |
|        | Conv. (13) | 79.8 | 81.4 | 92.6 | 87.9 | 89.4 | 84.3 | 45.5 | 89.5 |
| Trigram | DisC (2) | 80.0 | 81.3 | 92.6 | 87.9 | **90.0** | 85.2 | **46.7** | **89.6** |
|         | Conv. (13) | 79.7 | **81.6** | **92.8** | **87.9** | 89.8 | 84.2 | 45.8 | 89.5 |

Table 4: Performance comparison of element-wise product (DisC) and circular convolution for encoding local cooccurrences (best result for each task is **bolded**). Evaluation settings are the same as in Section 6. Note that for unigrams the two representations are equivalent.

# B  DOCUMENT EMBEDDINGS

## B.1  PERFORMANCE COMPARISON OF ALTERNATIVE REPRESENTATIONS

In this section we compare the performance of several alternative representations with the ones presented in the main evaluation (Table 1). Table 3 provides a numerical justification for our use of unordered $n$-grams (cooccurrences) instead of $n$-grams, as the performance of the two featurizations are closely comparable. In Table 4 we examine the use of circular convolution instead of elementwise multiplication as linear measurements of BonC vectors Plate (1995). To construct the former from a document $w_1, \ldots, w_T$ we compute

$$\left[ \sum_{t=1}^{T} v_{w_t}, \ldots, \sum_{t=1}^{T-n+1} \mathcal{F}^{-1} \left( \prod_{\tau=t}^{t+n-1} \mathcal{F}(v_{w_\tau}) \right) \right] \tag{13}$$

where $\mathcal{F}$ is the discrete Fourier transform and $\mathcal{F}^{-1}$ its inverse. Note that for $n = 1$ this is equivalent to the simple unigram embedding (and thus also to the DisC embedding in (2)).

## B.2  SCALING FACTOR FOR DISC EMBEDDINGS

**Lemma B.1.** *If word vectors $v_w \in \mathbb{R}^d$ are drawn i.i.d. from $\mathcal{U}^d\{\pm 1/\sqrt{d}\}$ then for any $n$-gram $g = (w_1, \ldots, w_n)$ we have $\mathbb{E}\|\tilde{v}_g\|_2^2 = 1$. The same result holds true with the additional assumption that all words in $g$ are distinct if the word vectors are i.i.d. $d$-dimensional spherical Gaussians.*

*Proof.*

$$\mathbb{E}\|\tilde{v}_g\|_2^2 = \sum_{i=1}^{d} \mathbb{E} \left( d^{\frac{n-1}{2}} \prod_{k=1}^{n} v_{w_k i} \right)^2 = d^{n-1} \sum_{i=1}^{d} \mathbb{E} \left( \prod_{k=1}^{n} v_{w_k i}^2 \right) = d^{n-1} \sum_{i=1}^{d} \prod_{i=1}^{n} \frac{1}{d} = 1$$

$\square$

### B.3 Proof of Proposition 3.1

Let $f(x) = i(x) = g(x) = x$ with

$$\mathcal{T}_f(v_{w_t}, h_{t-1}) = \begin{pmatrix} \mathbf{1}_{nd} \\ \mathbf{0}_{(n-1)d} \end{pmatrix}$$

$$\mathcal{T}_i(v_{w_t}, h_{t-1}) = \begin{pmatrix} \mathbf{0}_{d \times nd} & \cdots & & \mathbf{0}_{d \times d} \\ \vdots & & I_{(n-2)d} & \mathbf{0}_{(n-2)d \times d} \\ \vdots & & \ddots & I_d \\ \vdots & & \ddots & \mathbf{0}_{d \times d} \\ \mathbf{0}_{(n-2)d \times nd} & I_{(n-2)d} & \mathbf{0}_{(n-2)d \times d} \end{pmatrix} h_{t-1} + \begin{pmatrix} \mathbf{1}_d \\ \mathbf{0}_{(n-1)d} \\ \mathbf{1}_d \\ \mathbf{0}_{(n-2)d} \end{pmatrix}$$

$$\mathcal{T}_g(v_{w_t}, h_{t-1}) = \begin{pmatrix} C_1 I_d \\ \vdots \\ C_n d^{\frac{n-1}{2}} I_d \\ I_d \\ \vdots \\ I_d \end{pmatrix} v_{w_t}$$

Substituting these parameters into the LSTM update (3) and using $h_0 = 0$ we have $\forall\, t > 0$ that

$$h_t = \begin{pmatrix} C_1 \sum_{\tau=1}^{t} v_{w_\tau} \\ \vdots \\ C_n d^{\frac{n-1}{2}} \sum_{\tau=1}^{t-n+1} \bigodot_{k=1}^{n} v_{w_{\tau+k-1}} \\ v_{w_t} \\ \vdots \\ \bigodot_{k=1}^{n-1} v_{w_{t+k-n+1}} \end{pmatrix} = \begin{pmatrix} C_1 \sum_{\tau=1}^{t} \tilde{v}_{w_\tau} \\ \vdots \\ C_n d^{\frac{n-1}{2}} \sum_{\tau=1}^{t-n+1} \tilde{v}_{\{w_\tau, \ldots, w_{\tau+n-1}\}} \\ \tilde{v}_{w_t} \\ \vdots \\ \tilde{v}_{\{w_{t-n+2}, \ldots, w_t\}} \end{pmatrix}$$

Thus

$$h_T = \begin{pmatrix} C_1 \sum_{t=1}^{T} \tilde{v}_{w_t} \\ \vdots \\ C_n d^{\frac{n-1}{2}} \sum_{t=1}^{T-n+1} \tilde{v}_{\{w_t, \ldots, w_{t+n-1}\}} \\ \tilde{v}_{w_T} \\ \vdots \\ \tilde{v}_{\{w_{T-n+2}, \ldots, w_T\}} \end{pmatrix} = \begin{pmatrix} \tilde{z}^{(n)} \\ \tilde{v}_{w_T} \\ \vdots \\ \tilde{v}_{\{w_{T-n+2}, \ldots, w_T\}} \end{pmatrix}$$

Note that $h_t \in \mathbb{R}^{(2n-1)d}$ so as desired the LSTM has $\mathcal{O}(nd)$-memory. Although $h_T$ contains $(n-1)d$ more dimensions than $\tilde{z}^{(n)}$, by padding the end of the document with an end-of-document token whose word vector is $\mathbf{0}_d$ the entries in those dimensions will be set to zero by the update at the last step. Thus up to zero padding we will have $z^{\text{LSTM}} = h_T = \tilde{z}^{(n)}$.

## C Proof of Theorem 4.2

Throughout this section we assume the setting described in Theorem 4.2. Furthermore for some positive constant $C$ define the $\ell_2$-regularization of the loss function $\ell$ as

$$L(w) = \ell(w) + \frac{1}{2C}\|w\|_2^2$$

**Lemma C.1.** *Let $\hat{w}$ be the classifier obtained minimizing $L_S(w) = \frac{1}{m}\sum_{i=1}^{m}\ell(w^T x_i, y_i) + \frac{1}{2C}\|w\|_2^2$, where $\ell(\cdot, \cdot)$ is a convex $\lambda$-Lipschitz function in the first cordinate. Then*

$$\hat{w} = \sum_{i=1}^{m}\alpha_i y_i x_i \tag{14}$$

*where $|\alpha_i| \leq \frac{\lambda C}{m}\ \forall\ i$. This result holds in the compressed domain as well.*

*Proof.* If $\ell$ is an $\lambda$-Lipschitz function, its sub-gradient at every point is bounded by $\lambda$. So by convexity, the unique optimizer is given by taking first-order conditions:

$$0 = \partial_w L_S(w) = \frac{w}{C} + \frac{1}{m}\sum_{i=1}^{m}\partial_{w^T x_i}\ell(w^T x_i, y_i)x_i$$

$$\implies \hat{w} = \frac{C}{m}\sum_{i=1}^{m}-y_i\partial_{\hat{w}^T x_i}\ell(\hat{w}^T x_i, y_i)y_i x_i \tag{15}$$

Since $\ell$ is Lipschitz, $|\partial_{w^T x_i}\ell(w^T x_i, y_i)| \leq \lambda$. Therefore the first-order optimal solution (15) of $\hat{w}$ can be expressed as (14) for some $\alpha_1, \ldots, \alpha_m$ satisfying $|\alpha_i| \leq \frac{\lambda C}{m}\ \forall\ i$, which is the desired result.

$\square$

**Lemma C.2.** $x, x' \in \mathcal{X} \implies (1+\varepsilon)x^T x' - 2R^2\varepsilon \leq (Ax)^T(Ax') \leq (1-\varepsilon)x^T x' + 2R^2\varepsilon$

*Proof.* Since $A$ is $(\Delta\mathcal{X}, \varepsilon)$-RIP we have $(1-\varepsilon)\|x - x'\|_2 \leq \|A(x - x')\|_2 \leq (1+\varepsilon)\|x - x'\|_2$. Also since $\mathbf{0}_N \in \mathcal{X}$, $A$ is also $(\mathcal{X}, \varepsilon)$-RIP and the result then follows by the same argument as in (Calderbank et al., 2009, Lemma 4.2-3). $\square$

**Corollary C.1.** $\|\hat{w}\|_2^2 \leq \lambda^2 C^2 R^2$ *and* $\|\hat{w}_A\|_2^2 \leq \lambda^2 C^2(1+\varepsilon)^2 R^2$.

*Proof.* The first bound follows by expanding $\|\hat{w}\|_2^2$ and using $\|x\|_2 \leq R$; the second follows by expanding $\|\hat{w}_A\|_2^2$, applying Lemma C.2 to bound inner product distortion, and using $\|x\|_2 \leq R$. $\square$

**Lemma C.3.** *Let $\hat{w}$ be the linear classifier minimizing $L_S$. Then*

$$L_{\mathcal{D}}(A\hat{w}) \leq L_{\mathcal{D}}(\hat{w}) + \mathcal{O}(\lambda^2 C R^2 \varepsilon)$$

*Proof.* By Lemma C.1 we can re-express $\hat{w}$ using Equation 14 and then apply the inequality from Lemma C.2 to get

$$
\begin{aligned}
(A\hat{w})^T(Ax) &= \sum_{i=1}^{m}\alpha_i y_i (Ax_i)^T(Ax) \\
&\leq \sum_{i:\alpha_i y_i \geq 0}\alpha_i y_i \left((1-\varepsilon)x_i^T x + 2R^2\varepsilon\right) + \sum_{i:\alpha_i y_i < 0}\alpha_i y_i \left((1+\varepsilon)x_i^T x - 2R^2\varepsilon\right) \\
&= \hat{w}^T x - \varepsilon\sum_{i=1}^{m}|\alpha_i y_i|x_i^T x + 2R^2\varepsilon\sum_{i=1}^{m}|\alpha_i y_i| \leq \hat{w}^T x + 3\lambda C R^2\varepsilon
\end{aligned}
$$

$$
\begin{aligned}
(A\hat{w})^T(Ax) &= \sum_{i=1}^{m}\alpha_i y_i (Ax_i)^T(Ax) \\
&\geq \sum_{i:\alpha_i y_i \geq 0}\alpha_i y_i \left((1+\varepsilon)x_i^T x - 2R^2\varepsilon\right) + \sum_{i:\alpha_i y_i < 0}\alpha_i y_i \left((1-\varepsilon)x_i^T x + 2R^2\varepsilon\right) \\
&= \hat{w}^T x + \varepsilon\sum_{i=1}^{m}|\alpha_i y_i|x_i^T x - 2R^2\varepsilon\sum_{i=1}^{m}|\alpha_i y_i| \geq \hat{w}^T x - 3\lambda C R^2\varepsilon
\end{aligned}
$$

for any $x \in \mathbb{R}^N$. Since $\ell$ is $\lambda$-Lipschitz taking expectations over $\mathcal{D}$ implies

$$\ell_{\mathcal{D}}(A\hat{w}) \leq \ell_{\mathcal{D}}(\hat{w}) + 3\lambda^2 C R^2 \varepsilon \tag{16}$$

Substituting Equation 14 applying Lemma C.2 also yields

$$
\begin{aligned}
\|A\hat{w}\|_2^2 &= \sum_{i=1}^m \sum_{j=1}^m \alpha_i \alpha_j y_i y_j (A x_i)^T (A x_j) \\
&\leq \sum_{i,j:\alpha_i\alpha_j y_i y_j \geq 0} \alpha_i \alpha_j y_i y_j \left( (1-\varepsilon) x_i^T x_j + 2R^2\varepsilon \right) \\
&\quad + \sum_{i,j:\alpha_i\alpha_j y_i y_j < 0} \alpha_i \alpha_j y_i y_j \left( (1+\varepsilon) x_i^T x_j - 2R^2\varepsilon \right) \\
&\leq \sum_{i,j} \alpha_i \alpha_j y_i y_j x_i^T x_j + \sum_{i,j} -|\alpha_i\alpha_j y_i y_j|\varepsilon x_i^T x_j + 2R^2|\alpha_i\alpha_j y_i y_j|\varepsilon \\
&\leq \|\hat{w}\|_2^2 + 3\lambda^2 C^2 R^2 \varepsilon
\end{aligned}
$$

which implies

$$\frac{1}{2C}\|A\hat{w}\|_2^2 \leq \frac{1}{2C}\|\hat{w}\|_2^2 + \frac{3}{2}\lambda^2 C R^2 \varepsilon \tag{17}$$

Together the inequalities bounding the loss term (16) and the regularization term (17) imply the result.

$\square$

**Lemma C.4.** *Let $\hat{w}$ be the linear classifier minimizing $L_S$ and let $w^*$ be the linear classifier minimizing $L_{\mathcal{D}}$. Then with probability $1 - \gamma$*

$$L_{\mathcal{D}}(\hat{w}) \leq L_{\mathcal{D}}(w^*) + \mathcal{O}\left(\frac{\lambda^2 C R^2}{m} \log \frac{1}{\gamma}\right)$$

*This result holds in the compressed domain as well.*

*Proof.* By Corollary C.1 we have that $\hat{w}$ is contained in a closed convex subset independent of $S$. Therefore since $\ell$ is $\lambda$-Lipschitz, $L$ is $\frac{1}{C}$-strongly convex, and $\|x\|_2 \leq \mathcal{O}(R)$, we have by (Sridharan et al., 2008, Theorem 1) that with probability $1 - \gamma$

$$L_{\mathcal{D}}(\hat{w}) - L_{\mathcal{D}}(w^*) \leq 2\left[L_S(\hat{w}) - L_S(w^*)\right]_+ + \mathcal{O}\left(\frac{\lambda^2 C R^2}{m} \log \frac{1}{\gamma}\right)$$

Then since by definition $\hat{w}$ minimizes $L_S(w)$ we have that $L_S(\hat{w}) \leq L_S(w^*)$, which substituted into the previous equation completes the proof. $\square$

*Proof of Theorem 4.2.* Applying Lemma C.4 in the compressed domain yields

$$\ell_{\mathcal{D}}(\hat{w}_A) \leq \ell_{\mathcal{D}}(\hat{w}_A) + \frac{1}{2C}\|\hat{w}_A\|_2^2 = L_{\mathcal{D}}(\hat{w}_A) \leq L_{\mathcal{D}}(w_A^*) + \mathcal{O}\left(\frac{\lambda^2 C R^2}{m} \log \frac{1}{\gamma}\right)$$

where $w_A^*$ minimizes $L_{\mathcal{D}}$. By definition of $w_A^*$, $L_{\mathcal{D}}(w_A^*) \leq L_{\mathcal{D}}(A\hat{w})$, so together with Lemma C.3 and the previous inequality we have

$$\ell_{\mathcal{D}}(\hat{w}_A) \leq L_{\mathcal{D}}(A\hat{w}) + \mathcal{O}\left(\frac{\lambda^2 C R^2}{m} \log \frac{1}{\gamma}\right) \leq L_{\mathcal{D}}(\hat{w}) + \mathcal{O}\left(\lambda^2 C R^2 \left(\varepsilon + \frac{1}{m}\log\frac{1}{\gamma}\right)\right)$$

We now apply Lemma C.4 in the sparse domain to get

$$\ell_{\mathcal{D}}(\hat{w}_A) \leq L_{\mathcal{D}}(w^*) + \mathcal{O}\left(\lambda^2 C R^2 \left(\varepsilon + \frac{1}{m}\log\frac{1}{\gamma}\right)\right)$$

where $w^*$ minimizes $L_{\mathcal{D}}$. By definition of $w^*$, $L_{\mathcal{D}}(w^*) \leq L_{\mathcal{D}}(w_0) = \ell_{\mathcal{D}}(w_0) + \frac{1}{2C}\|w_0\|_2^2$, so by the previous inequality we have

$$\ell_{\mathcal{D}}(\hat{w}_A) \leq \ell_{\mathcal{D}}(w_0) + \frac{1}{2C}\|w_0\|_2^2 + \mathcal{O}\left(\lambda^2 C R^2 \left(\varepsilon + \frac{1}{m}\log\frac{1}{\gamma}\right)\right)$$

Substituting the $C$ that minimizes the r.h.s. of this inequality completes the proof. $\square$

## D    PROOF OF LEMMA 4.1

We assume the setting described in Lemma 4.1, where we are concerned with the RIP condition of the matrix $A^{(n)}$ when multiplying vectors $x \in \mathcal{X}_T^{(n)}$, the set of BonC vectors for documents of length at most $T$. This matrix can be written as

$$A^{(n)} = \begin{pmatrix} A_1 & \mathbf{0}_{d \times V_2} & \cdots & \mathbf{0}_{d \times V_n} \\ \mathbf{0}_{d \times V_1} & \ddots & \ddots & \vdots \\ \vdots & \ddots & \ddots & \mathbf{0}_{d \times V_n} \\ \mathbf{0}_{d \times V_1} & \cdots & \mathbf{0}_{d \times V_{n-1}} & A_n \end{pmatrix}$$

where $A_p$ is the $d \times V_p$ matrix whose columns are the DisC embeddings of all $p$-grams in the vocabulary (and thus $A^{(1)} = A_1 = A$, the matrix of the original word embeddings). Note that from (1) any $x \in \mathcal{X}_T^{(n)}$ can be written as $x = [x_1, \ldots, x_n]$, where $x_p$ is a $T$-sparse vector whose entries correspond to $p$-grams. Thus we also have $A^{(n)}x = [A_1 x_1, \ldots, A_n x_n]$.

**Lemma D.1.** *If $A_p$ is $(2k, \varepsilon)$-RIP w.p. $1 - \gamma \, \forall \, p \in [n]$ then $A^{(n)}$ is $\left( \Delta \mathcal{X}_k^{(n)}, \varepsilon \right)$-RIP w.p. at least $1 - n\gamma$.*

*Proof.* By union bound we have that $A_p$ is $(2k, \varepsilon)$-RIP $\forall \, p \in [n]$ with probability at least $1 - n\gamma$. Thus by Definition 4.1 we have w.p. $1 - n\gamma$ that $\forall \, x \in \Delta \mathcal{X}_k^{(n)}$

$$\|A^{(n)}x\|_2^2 = \sum_{p=1}^n \|A_p x_p\|_2^2 \leq \sum_{p=1}^n (1 + \varepsilon)^2 \|x_p\|_2^2 = (1 + \varepsilon)^2 \|x\|_2^2$$

Similarly, $\|A^{(n)}x\|_2^2 \geq (1 - \varepsilon)^2 \|x\|_2^2$. From Definition 4.1, taking the square root of both sides of both inequalities completes the proof. □

**Definition D.1** (Foucart & Rauhut, 2013). *Let $\mathcal{D}$ be a distribution over a subset $S \subset \mathbb{R}^n$. Then the set $\Phi = \{\phi_1, \ldots, \phi_N\}$ of functions $\phi_i : S \mapsto \mathbb{R}$ is a bounded orthonormal system (BOS) with constant $B$ if we have $\mathbb{E}_{\mathcal{D}}(\phi_i \phi_j) = 1_{i=j} \, \forall \, i, j$ and $\sup_{s \in S} |\phi_i(s)| \leq B \, \forall \, i$. Note that by definition $B \geq 1$.*

**Theorem D.1** (Foucart & Rauhut, 2013). *If $d = \tilde{\Omega}\left( \frac{B^2 k}{\varepsilon^2} \log \frac{N}{\gamma} \right)$ for $(\varepsilon, \gamma) \in (0, 1)$ and $\sqrt{d}A$ is a $d \times N$ matrix associated with a BOS with constant $B$ then $A$ is $(k, \varepsilon)$-RIP w.p. $1 - \gamma$.*

**Lemma D.2.** *If $d = \tilde{\Omega}\left( \frac{T}{\varepsilon^2} \log \frac{V_p}{\gamma} \right)$ and the word embeddings are drawn i.i.d. from $\mathcal{U}^d\{\pm 1/\sqrt{d}\}$ then for any $p \in [n]$ the matrix $A_p \in \mathbb{R}^{d \times V_p}$ of DisC embeddings is $(T, \varepsilon)$-RIP w.p. $1 - \gamma$.*

*Proof.* Note that by Theorem D.1 it suffices to show that $\sqrt{d}A_p$ is a random sampling matrix associated with a BOS with constant $B = 1$. Let $\mathcal{D} = \mathcal{U}^V\{\pm 1\}$ be the uniform distribution over $V$ i.i.d. Rademacher random variables indexed by words in the vocabulary. Then by definition the matrix $A_p \in \mathbb{R}^{d \times V_p}$ can be constructed by drawing random variables $x^{(1)}, \ldots, x^{(d)}$ i.i.d. from $\mathcal{D}$ and assigning to the $ij$th entry of $\sqrt{d}A_p$ corresponding to the $p$-gram $g = \{g_1, \ldots, g_p\}$ the value $\phi_j\left(x^{(i)}\right) = \prod_{t=1}^p x_{g_t}^{(i)}$, where each function $\phi_j : \{\pm 1\}^V \mapsto \mathbb{R}$ is uniquely associated to its $p$-gram. It remains to be shown that this set of functions is a BOS with constant $B = 1$.

For any two $p$-grams $g, g'$ and their functions $\phi_i, \phi_j$ we have $\mathbb{E}_{\mathcal{D}}(\phi_i \phi_j) = \mathbb{E}_{x \sim \mathcal{D}}\left( \prod_{t=1}^p x_{g_t} x_{g'_t} \right)$, which will be 1 iff each word in $g \cup g'$ occurs an even number of times in the product and 0 otherwise. Because all $p$-grams are uniquely defined under any permutation of its words (i.e. we are in fact using $p$-cooccurrences) and we have assumed that no $p$-gram contains a word more than once, each word occurs an even number of times in the product iff $g = g' \iff i = j$. Furthermore we have that $|\phi_i(x)| \leq 1 \, \forall \, x \in \{\pm 1\}^V \, \forall \, i$ by construction. Thus according to Definition D.1 the set of functions $\{\phi_1, \ldots, \phi_{V_p}\}$ associated to the $p$-grams in the vocabulary is a BOS with constant $B = 1$. □

*Proof of Lemma 4.1.* Since $d = \tilde{\Omega}\left( \frac{T}{\varepsilon^2} \log \frac{n V_n^{\max}}{\gamma} \right)$, Lemma D.2 implies that $A_p$ is $(2T, \varepsilon)$-RIP w.p. $1 - \frac{\gamma}{n} \, \forall \, p \in [n]$. Applying Lemma D.1 yields the result. □

# E   DETAILS OF THE SUPPORTING HYPERPLANE PROPERTY

In Section 5.2, Definition 5.1 we introduced the Supporting Hyperplane Property (SHP), which by Corollary 5.1 characterizes when BP+ perfectly recovers a nonnegative signal. Together with Lemmas A.1 and A.2 this fact also shows that SHP is a weaker condition than the well-known nullspace property (NSP):

**Corollary E.1.** *If a matrix $A \in \mathbb{R}^{d \times N}$ with columns in general position satisfies NSP for some $S \subset [N]$ then it also satisfies S-SHP.*

In this section we give the entire proof of Corollary 5.1 and describe how to verify SHP given a design matrix and a set of support indices.

## E.1   PROOF OF COROLLARY 5.1

Recall that it suffices to show equivalence of $A$ being $S$-SHP with the columns $A_S$ forming the vertices of a $k$-dimensional face of $\mathrm{conv}(A)$, where we can abuse notation to set $A \in \mathbb{R}^{d \times (N+1)}$, with the extra column being the origin $\mathbf{0}_d$, so long as we constrain $N + 1 \notin S$.

( $\Longrightarrow$ ): The proof of the forward direction appeared in full in the proof sketch (see Section 5.2).

( $\Longleftarrow$ ): A subset $F \subset \mathbb{R}^d$ is a face of $\mathrm{conv}(A)$ if for some hyperplane $H = \{v : a^T v - b = 0\}$ we have $F = \mathrm{conv}(A) \cap H$ and $\mathrm{conv}(A) \backslash F \subseteq H_- = \{v : a^T v - b < 0\}$, where $H_-$ is the negative halfspace of $H$. Define the simplex $\Delta_m = \{\lambda \in [0,1]^m : \sum_{i=1}^m \lambda_i = 1\}$.

Since $A$ is $S$-SHP we have a hyperplane $H = \{v : a^T v - b = 0\}$ containing the columns $A_S$ such that $A_{\overline{S}} \subset H_-$. Thus $a^T A_i - b = 0 \; \forall \, i \in S$ and $a^T A_i - b < 0 \; \forall \, i \notin S$. We also know that $F = \{\sum_{i \in S} \lambda_i A_i : \lambda \in \Delta_{|S|}\} \subseteq H$ by convexity of $H$. Since any point $y \in \mathrm{conv}(A) \backslash F$ can be written as $y = \sum_{i=1}^{N+1} \lambda_i A_i$ for some $\lambda \in \Delta_{N+1}$ such that $\exists \, j \notin S$ such that $\lambda_j \neq 0$, we have that

$$a^T y - b = \sum_{i \in S} \lambda_i (a^T A_i - b) + \sum_{j \notin S} \lambda_j (a^T A_j - b) = \sum_{j \notin S} \lambda_j (a^T A_j - b) < 0$$

This implies that $\mathrm{conv}(A) \backslash F \subseteq H_-$ and $F = \mathrm{conv}(A) \cap H$, so since the columns of $A$ are in general position $F$ is a $k$-dimensional face of $\mathrm{conv}(A)$ whose vertices are the columns $A_S$.

## E.2   VERIFYING SHP

Recall that a matrix $\mathbb{R}^{d \times N}$ satisfies $S$-SHP for $S \subset [N]$ if there is a hyperplane containing the set of all columns of $A$ indexed by $S$ and the set of all other columns together with the origin are on one side of it. Due to Corollary 5.1, checking $S$-SHP allows us to know whether all nonnegative signals with index support $S$ will be recovered by BP+ without actually running the optimization on any one of them. The property can be checked by solving a convex problem of the form

$$\min_{h \in \mathbb{R}^{d+1}} \sum_{i \notin S} \max \left\{ \tilde{A}_i^T h + \varepsilon, 0 \right\}^p \quad \text{subject to} \quad \tilde{A}_S^T h = \mathbf{0}_{|S|}$$

$$\text{where} \quad \tilde{A} = \begin{pmatrix} A & \mathbf{0}_d \\ \mathbf{1}_N^T & 1 \end{pmatrix} \quad \text{and} \quad \varepsilon > 0, \; p \geq 1$$

The constraint enforces the property that the hyperplane contains all support embeddings, while the optimal objective value is zero iff SHP is satisfied (this follows from the fact that scaling $h$ does not affect the constraint so if the minimal objective is zero for any single $\varepsilon > 0$ it is zero for all $\varepsilon > 0$). The problem can be solved via using standard first or second-order equality-constrained convex optimization algorithms. We set $\varepsilon = 1$ and $p = 3$ (to get a $\mathcal{C}^2$ objective) and adapt the second-order method from Boyd & Vandenberghe (2004, Chapter 10). Our implementation can be found at `https://github.com/NLPrinceton/sparse_recovery`.

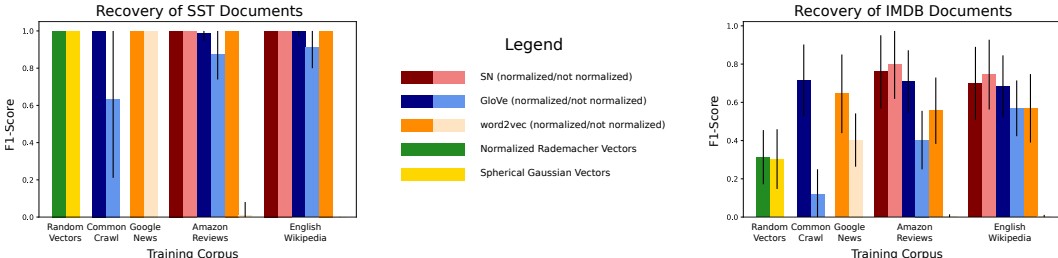

Figure 7: Efficiency of pretrained embeddings as sensing vectors at $d = 300$ dimensions, measured via the $F_1$-score of the original BoW. 200 documents from each dataset were compressed and recovered in this experiment. For fairness, the number of words $V$ is the same for all embeddings so all documents are required to be subsets of the vocabulary of all corpora. word2vec embeddings trained on Google News and GloVe vectors trained on Common Crawl were obtained from public repositories (Mikolov et al., 2013; Pennington et al., 2014) while Amazon and Wikipedia embeddings were trained for 100 iterations using a symmetric window of size 10, a min count of 100, for SN/GloVe a cooccurrence cutoff of 1000, and for word2vec a down-sampling frequency cutoff of $10^{-5}$ and a negative example setting of 3. 300-dimensional normalized random vectors are used as a baseline.

## F  SPARSE RECOVERY WITH PRETRAINED EMBEDDINGS

### F.1  PERFORMANCE COMPARISON OF ALTERNATIVE EMBEDDINGS

We show in Figure 7 that the surprising effectiveness of word embeddings as linear measurement vectors for BoW signals holds for other embedding objectives and corpora as well. Specifically, we see that widely used embeddings, when normalized, match the efficiency of random vectors for retrieving SST BoW and are more efficient when retrieving IMDB BoW. Interestingly, SN vectors are most efficient and are also the only embeddings for normalizing is not needed for good performance.

### F.2  A MODEL-BASED THEORETICAL APPROACH

In Section 5.2 we gave some intuition for why pretrained word embeddings are efficient sensing vectors for natural language BoW by examining a geometric characterization of local equivalence due to Donoho & Tanner (2005) in light of the usual similarity properties of word embeddings. However, this analysis does not provide a rigorous theory for our empirical results. In this section we briefly discuss a model-based justification that may lead to a stronger understanding.

We need a model relating BoW generation to the word embeddings trained over words co-occurring in the same BoW. As a starting point consider the model of Arora et al. (2016), in which a corpus is generated by a random walk $c_t$ over the surface of a ball in $\mathbb{R}^d$; at each $t$ a word $w$ is emitted w.p.

$$\mathbb{P}(w|c_t) \propto \exp\langle c_t, v_w \rangle \tag{18}$$

Minimizing the SN objective approximately maximizes corpus likelihood under this model.

Thus in an approximate sense a document of length $T$ is generated by setting a *context vector* $c$ and emitting $T$ words via (18) with $c_t = c$. This model is a convenient one for analysis due its simplicity and invariance to word order as well as the fact that the approximate maximum likelihood document vector is the sum of the embeddings of words in the document. Building upon the intuition established following Corollary 5.1 one can argue that, if we have the true latent SN vectors, then embeddings of words in the same document (i.e. emitted by the same context vector) will be close to each other and thus easy to separate from the embeddings of other words in the vocabulary.

However, we find empirically that not all of the $T$ words closest to the sum of the word embeddings (i.e. the context vector) are the ones emitted; indeed individual word vectors in a document may have small, even negative inner product with the context vector and still be recovered via BP. Thus any further theoretical argument must also be able to handle the recovery of lower probability words whose vectors are further away from the context vector than those of words that do not appear in the document. We thus leave to future work the challenge of explaining why embeddings resulting from this (or another) model provide such efficient sensing matrices for natural language BoW.

