# OpenReview forum: "A Compressed Sensing View of Unsupervised Text Embeddings, Bag-of-n-Grams, and LSTMs"
_ICLR.cc/2018/Conference — Accept (Poster)_

### Official Review · AnonReviewer2 · 2017-11-26
**The paper applies techniques from compressed sensing to analyze the classification performance of LSTM word embeddings**

**Rating:** 7
**Confidence:** 3

**Review:**

The main insight in this paper is that LSTMs can be viewed as producing a sort of sketch of tensor representations of n-grams.  This allows the authors to design a matrix that maps bag-of-n-gram embeddings into the LSTM embeddings. They then show that the result matrix satisfies a restricted isometry condition.  Combining these results allows them to argue that the classification performance based on LSTM embeddings is comparable to that based on bag-of-n-gram embeddings.

I didn't check all the proof details, but based on my knowledge of compressed sensing theory, the results seem plausible. I think the paper is a nice contribution to the theoretical analysis of LSTM word embeddings.

---

> ### Author Response · Authors · 2017-12-11
> **Response to AnonReviewer2**
>
> Thank you for the positive review! We are currently preparing a revision incorporating these comments. We would also like to clarify that our paper concerns LSTM document embeddings, not word embeddings.

---

### Official Review · AnonReviewer1 · 2017-11-27
**Interesting paper**

**Rating:** 7
**Confidence:** 1

**Review:**

The interesting paper provides theoretical support for the low-dimensional vector embeddings computed using LSTMs or simple techniques, using tools from compressed sensing. The paper also provides numerical results to support their theoretical findings. The paper is well presented and organized.

-In theorem 4.1, the embedding dimension $d$ is depending on $T^2$, and it may scale poorly with respect to $T$.

---

> ### Author Response · Authors · 2017-12-11
> **Response to AnonReviewer1**
>
> Thank you for the positive review! We are currently preparing a revision incorporating these comments.
>
> Comment: “the embedding dimension $d$ is depending on $T^2$, and it may scale poorly with respect to $T$.”
> Yes the bound may scale poorly with document length. At the moment many tasks in this area use short sentences (e.g. SST has avg. length < 20), and Fig. 4 indicates convergence of DisC to BonC performance even on the IMDB task (avg. length > 250) so perhaps our bound is too pessimistic. Note that in the unigram (BoW) case the scaling is (provably) linear in T because then the design matrix is an i.i.d. Rademacher ensemble.

---

### Official Review · AnonReviewer3 · 2017-11-28
**The paper studies text embeddings through the lens of compressive sensing theory. The authors proved that, for the proposed embedding scheme, certain LSTMs with random initialization are at least as good as the linear classifiers; the theorem is almost a direction application of the RIP of random Rademacher matrices.**

**Rating:** 6
**Confidence:** 4

**Review:**

My review reflects more from the compressive sensing perspective, instead that of deep learners.

In general, I find many of the observations in this paper interesting. However, this paper is not strong enough as a theory paper; rather, the value lies perhaps in its fresh perspective.

The paper studies text embeddings through the lens of compressive sensing theory. The authors proved that, for the proposed embedding scheme, certain LSTMs with random initialization are at least as good as the linear classifiers; the theorem is almost a direction application of the RIP of random Rademacher matrices. Several simplifying assumptions are introduced, which rendered the implication of the main theorem vague, but it can serve as a good start for the hardcore statistical learning-theoretical analysis to follow.

The second contribution of the paper is the (empirical) observation that, in terms of sparse recovery of embedded words, the pretrained embeddings are better than random matrices, the latter being the main focus of compressive sensing theory. Partial explanations are provided, again using results in compressive sensing theory. In my personal opinion, the explanations are opaque and unsatisfactory. An alternative route is suggested in my detailed review.
Finally, extensive experiments are conducted and they are in accordance with the theory.

My most criticism regarding this paper is the narrow scope on compressive sensing, and this really undermines the potential contribution in Section 5.

Specifically, the authors considered only Basis Pursuit estimators for sparse recovery, and they used the RIP of design matrices as the main tool to argue what is explainable by compressive sensing and what is not. This seems to be somewhat of a tunnel-visioning for me: There are a variety of estimators in sparse recovery problems, and there are much less restrictive conditions than RIP of the design matrices that guarantee perfect recovery.

In particular, in Section 5, instead of invoking [Donoho&Tanner 2005], I believe that a more plausible approach is through [Chandrasekaran et al. 2012]. There, a simple deterministic condition (the null space property) for successful recovery is proved. It would be of direct interest to check whether such condition holds for a pretrained embedding (say GloVe) given some BoWs. Furthermore, it is proved in the same paper that Restricted Strong Convexity (RSC) alone is enough to guarantee successful recovery; RIP is not required at all. While, as the authors argued in Section 5.2, it is easy to see that pretrained embeddings can never possess RIP, they do not rule out the possibility of RSC.

Exactly the same comments above apply to many other common estimators (lasso, Dantzig selector, etc.) in compressive sensing which might be more tolerant to noise.

Several minor comments:

1. Please avoid the use of “information theory”, especially “classical information theory”, in the current context. These words should be reserved to studies of Channel Capacity/Source Coding `a la Shannon. I understand that in recent years people are expanding the realm of information theory, but as compressive sensing is a fascinating field that deserves its own name, there’s no need to mention information theory here.

2. In Theorem 4.1, please be specific about how the l2-regularization is chosen.

3. In Section 4.1, please briefly describe why you need to extend previous analysis to the Lipschitz case. I understood the necessity only through reading proofs.

4. Can the authors briefly comment on the two assumptions in Section 4, especially the second one (on n- cooccurrence)? Is this practical?

5. Page 1, there is a typo in the sentence preceding [Radfors et al., 2017].

6. Page 2, first paragraph of related work, the sentence “Our method also closely related to ...” is incomplete.

7. Page 2, second paragraph of related work, “Pagliardini also introduceD a linear ...”

8. Page 9, conclusion, the beginning sentence of the second paragraph is erroneous.

[1] Venkat Chandrasekaran, Benjamin Recht, Pablo A. Parrilo, Alan S. Willsky, “The Convex Geometry of Linear Inverse Problems”, Foundations of Computational Mathematics, 2012.

---

> ### Author Response · Authors · 2017-12-11
> **Response to AnonReviewer3**
>
> Thank you for the thorough review! We’ll revise incorporating your comments.
>
> Main Responses:
>
> 1) “instead of using [Donoho & Tanner 2005] it should be better to use [Chandrasekaran et al. 2012]’s deterministic condition, the null space property or NSP” (paraphrase)
>
> We knew of NSP but turned to Donoho & Tanner (2005) because NSP is difficult to work with (no obvious method to check if local NSP holds; checking global NSP is NP-hard (Tillmann & Pfetsch, 2014)). Since NSP is equivalent to exact recovery, our experiments (Fig. 1-2) strongly suggest that local NSP holds, but we did not find a way to use it to gain intuition or proofs. While closely related to NSP, the polytope condition of Donoho & Tanner (2005) implies Corollary 5.1, which suggests both a nice property of word embeddings and an efficient method to check recovery of nonnegative signals.
>
>  2): “[the claim that] certain LSTMs with random initialization are at least as good as the linear classifiers… ...is almost a direction application of the RIP of random Rademacher matrices”
>
> This is true for the unigram (BoW) case. The proof for the n-gram case necessitated constructing a design matrix with correlated entries for which RIP is not as obvious. We agree that the bigger technical contribution is in connecting these ideas to text embeddings.
>
>
> Other Responses:
>
> Restricted Strong Convexity (RSC): “it is proved in [Chandrasekran et al. 2012]  that Restricted Strong Convexity (RSC) alone is enough to guarantee successful recovery.”
>
> To our knowledge RSC is used mostly for the case of signal/measurement noise (Negahban et al., 2010; Chandrasekaran et al., 2012), whereas we are in the noiseless setting. We know of work by Elenberg et al. (2016) using RSC to guarantee recovery with Orthogonal Matching Pursuit, but we have found that such greedy methods do not work well for pretrained embeddings (Section 5.1 paragraph 2), indicating that a sufficient RSC condition does not hold.
>
> LASSO/Dantzig Selector: “the same comments above apply to many other common estimators (lasso, Dantzig selector, etc.) in compressive sensing which might be more tolerant to noise.”
>
> LASSO was in fact the first approach we tried, with similar results as Basis Pursuit (we refer to it in Section 5.1 paragraph 2 as an “l_0-surrogate method”). However, as we are in the noiseless setting we do not need the robustness provided by LASSO; indeed, experiments show it performs somewhat worse for both pretrained and random vectors. Furthermore, to our knowledge guarantees for LASSO often have analogous results for Basis Pursuit, so the theoretical benefit to studying it is unclear. Although we did not try the Dantzig Selector, it can also be seen as a robust extension of Basis Pursuit and so similarly does not provide a clear advantage in our case.
>
>
> Minor Points:
>
> 1. We use the phrase “classical information theory” only in connection with the scheme in Paskov et al., (2013) which is inspired by the Lempel-Ziv compression algorithm (Ziv & Lempel, 1977); 40 years old and directly inspired by Shannon’s works!
> 2. In theory the regularization constant C is chosen to minimize the error bound; in practice it is chosen by cross-validation.
> 3. We extend the analysis in order to handle logistic loss as it is commonly used in the NLP community and by supervised LSTMs. We do not need Theorem 4.2 to hold for all Lipschitz functions to get Theorem 4.1, but the function does need to be Lipschitz to control the error.
> 4.1 This assumption is without loss of generality and is made to remove a spurious dependence on T in the error bound.
> 4.2. There will sometimes be n-cooccurrences that contain a word more than once, e.g. (as, long, as), but they occur infrequently and can be removed by merging words as a preprocessing step. In the SST training corpus only 0.019% of bigrams and 0.75% of trigrams have this issue, the latter often due to words between two commas in a list.
> 5-8. Will be addressed in revision.
>
>
> V. Chandrasekaran, B. Recht, P. A. Parrilo, and A. S. Willsky. “The Convex Geometry of Linear Inverse Problems.” Found. of Comp. Mathematics 2012.
> D. L. Donoho and J. Tanner. “Sparse nonnegative solution of underdetermined linear equations by linear programming.” PNAS 2005.
> E. R. Elenberg, R. Khanna, A. G. Dimakis, and S. Negahban. “Restricted strong convexity implies weak submodularity.” arXiv 2016.
> S. Negahban, B. Yu, M. J. Wainwright, and P. K. Ravikumar. “A unified framework for high-dimensional analysis of M-estimators with decomposable regularizers.” NIPS 2009.
> H. S. Paskov, R. West, J. C. Mitchell, and T. J. Hastie. “Compressive feature learning.” NIPS 2013.
> A. M. Tillmann and M. E. Pfetsch. “The computational complexity of the restricted isometry property, the nullspace property, and related concepts in compressed sensing.” IEEE Trans. on Info. Theory 2014.
> J. Ziv and A. Lempel. “A Universal Algorithm for Sequential Data Compression.” IEEE Trans. on Info. Theory 1977.

---

> > ### Comment · AnonReviewer3 · 2017-12-12
> > **follow up**
> >
> > Thanks for the authors' detailed response. It seems that the authors missed some of my points, hence a few further clarifications below:
> >
> > 1) Regarding the conditions that I posed (NSP, RSC, etc.), my main point is that there exist much weaker conditions than RIP which are sufficient for sparse recovery. This should at least be explicitly mentioned, otherwise researchers in compressive sensing will have doubts to many of the statements in this paper.
> >
> > For instance, in Section 5, the authors wrote "These embeddings cannot fit into the usual compressed sensing worldview since the matrix defined by the embeddings cannot satisfy RIP." Compressed sensing is not RIP. Moreover, this is certainly not surprising to most high-dimensional statisticians as RIP is a very strong condition. On the other hand, they would be much more surprised if the RSC fails to hold (which is suggested by the authors in the response; I'm much more interested in this fact).
> >
> > 2) I did not ask the authors to verify the NSP, which is of course NP-hard, but rather to check empirically the conditions that imply NSP. For instance, the famous Restricted Singular/Eigenvalue Condition does the job (see Theorem 3.2 of [Chandrasekran et al. 2012]): Generate random directions in the tangent cone, compute their norms, and check the RSP constant distribution; notice that in this case RSC is equivalent to RS/EC. Again, it is very intriguing to me that the RSC fails not hold, as suggested by the authors.
> >
> > 3) LASSO/Dantzig selector are common compressive sensing algorithms (LASSO is possibly more popular than Basis Pursuit). Hence it feels weird to equate compressive sensing to BP in Appendix A. Maybe the authors can simply mention that this paper focuses on the BP algorithm, and defer the others to future work.
> >
> >
> > Some minor comments:
> >
> > 4) In the response the authors mentioned that they had tried LASSO but it didn't work as well. I'm not sure what the authors exactly did, but I guess in this case the *constrained* LASSO would perform well. Maybe the authors can take a shot.
> >
> > 5) The connection to "classical information theory" is still quite vague (you refer to a paper which was inspired by the MDL and used LZ77, and I see very little connection with this paper). I personally feel that using "classical information theory" only damages the credibility of this paper to the experts. The simplest way to remedy this is to remove the term; otherwise, making the connection explicit is also an option.

---

> > > ### Author Response · Authors · 2017-12-20
> > > **Response to follow up**
> > >
> > > Thank you for following up.
> > >
> > > 1) “Much weaker conditions than RIP that are sufficient for sparse recovery should be explicitly mentioned. The authors say "These embeddings cannot fit into the usual compressed sensing worldview since the matrix defined by the embeddings cannot satisfy RIP." Compressed sensing is not RIP.” (Paraphrase)
> > >
> > > You are right; as discussed later in the paper an explanation should likely come from some other (weaker) condition. We chose to focus on the polytope condition but agree that discussion of others (like REC) is warranted and will include it in revision.
> > >
> > >
> > > 2) “Rather than verify NSP (NP-hard), empirically check conditions that imply it. For example, the restricted singular/eigenvalue condition (Theorem 3.2 of Chandrasekaran et al.) implies it. Try generating random directions in the tangent cone, computing their norms, and checking the constant distribution; notice that in this case RSC is equivalent to RS/EC.” (Paraphrase)
> > >
> > > We computed such upper bounds on RE constants** for both pretrained and random embeddings and found that they were smaller for the former (note that larger constants are better for recovery). Unfortunately this does not settle the issue about RE property for pretrained embeddings as lower bounds for random vectors are vacuous when recovery isn’t perfect (e.g. when the dimension is small enough that pretrained embeddings do better) (Banerjee et al., 2014). Note that verifying RE is also NP-hard (Dobriban & Fan, 2016). We will discuss these points in revision.
> > >
> > >
> > > (** We don’t know how to sample directions uniformly from tangent cone ---rejection sampling doesn’t work well as the intersection of the cone with the unit sphere is very small compared to the unit sphere--- so these bounds were computed in a greedy manner.)
> > >
> > >
> > > 3) “LASSO/Dantzig selector are common compressive sensing algorithms (LASSO is possibly more popular than Basis Pursuit). Hence it feels weird to equate compressive sensing to BP in Appendix A. Maybe the authors can simply mention that this paper focuses on the BP algorithm, and defer the others to future work.”
> > >
> > > Appendix A represents a brief overview of only the parts of compressed sensing needed in the main paper. We didn’t include LASSO/Dantzig because the word embeddings setting has no signal/measurement noise, in which case both methods are equivalent to BP. The revision will clarify.
> > >
> > >
> > > 4) “In the response the authors mentioned that they had tried LASSO but it didn't work as well. I'm not sure what the authors exactly did, but I guess in this case the *constrained* LASSO would perform well.”
> > >
> > > We have indeed tried constrained LASSO and it works quite well, but BP works slightly better. This does not seem surprising to us since we are in the noiseless setting. We’ll add a note to that effect.
> > >
> > >
> > > 5) “The connection to "classical information theory" is still quite vague (you refer to a paper which was inspired by the MDL and used LZ77, and I see very little connection with this paper).”
> > >
> > > We’re happy to omit the problematic phrase; it only refers to a past work of Paskov et al. (which uses compression ideas on Bag-of-n-Grams representation).
> > >
> > >
> > > A. Banerjee, S. Chen, F. Fazayeli, and V. Sivakumar. “Estimation with Norm Regularization.” NIPS 2014.
> > > E. Dobriban and J. Fan. “Regularity Properties for Sparse Regression.” Communications in Mathematics and Statistics 2016.

---

### Author Response · Authors · 2018-01-04
**Revision Summary**

Dear Readers,

We have uploaded a revision. There are two main changes:

1. An improvement to Lemma 4.1 that allows the embedding dimension $d$ in Theorem 4.1 to depend linearly (up to log factors) on the document length $T$. This directly addresses the concern of AnonReviewer1 that $d$ scales quadratically with $T$.

2. Updates to Section 5 and the Appendix to a) clarify why basis pursuit is the natural choice for our setting and b) discuss weak sparse recovery conditions (including NSP/REP) in greater depth to see how they can help understand why BoW recovery improves when the sensing matrix consists of word embeddings. We hope these changes address the concern of AnonReviewer3 about the scope of compressed sensing considered in the paper.

---

### Decision · Program_Chairs · 2018-01-29
**ICLR 2018 Conference Acceptance Decision**

**Decision:**

Accept (Poster)

**Comment:**

sadly, none of the reviewers seem to have been able to fully appreciate and check the proofs.

but in the words of even the least positive reviewer:
In general, I find many of the observations in this paper interesting. However, this paper is not strong enough as a theory paper; rather, the value lies perhaps in its fresh perspective.

i think we can all gain from fresh perspectives of LSTMs and DL for NLP :)